

# Optimizing the use of a sensor resource for opponent polarization coding

Francisco J.H. Heras[1,2] and Simon B. Laughlin[1]

[1] Department of Zoology, University of Cambridge, Cambridge, United Kingdom
[2] Current affiliation: Champalimaud Neuroscience Programme (CNP), Champalimaud Centre for the Unknown, Lisboa, Portugal

## ABSTRACT

Flies use specialized photoreceptors R7 and R8 in the dorsal rim area (DRA) to detect skylight polarization. R7 and R8 form a tiered waveguide (central rhabdomere pair, CRP) with R7 on top, filtering light delivered to R8. We examine how the division of a given resource, CRP length, between R7 and R8 affects their ability to code polarization angle. We model optical absorption to show how the length fractions allotted to R7 and R8 determine the rates at which they transduce photons, and correct these rates for transduction unit saturation. The rates give polarization signal and photon noise in R7, and in R8. Their signals are combined in an opponent unit, intrinsic noise added, and the unit's output analysed to extract two measures of coding ability, number of discriminable polarization angles and mutual information. A very long R7 maximizes opponent signal amplitude, but codes inefficiently due to photon noise in the very short R8. Discriminability and mutual information are optimized by maximizing signal to noise ratio, SNR. At lower light levels approximately equal lengths of R7 and R8 are optimal because photon noise dominates. At higher light levels intrinsic noise comes to dominate and a shorter R8 is optimum. The optimum R8 length fractions falls to one third. This intensity dependent range of optimal length fractions corresponds to the range observed in different fly species and is not affected by transduction unit saturation. We conclude that a limited resource, rhabdom length, can be divided between two polarization sensors, R7 and R8, to optimize opponent coding. We also find that coding ability increases sub-linearly with total rhabdom length, according to the law of diminishing returns. Consequently, the specialized shorter central rhabdom in the DRA codes polarization twice as efficiently with respect to rhabdom length than the longer rhabdom used in the rest of the eye.

## INTRODUCTION

Sunlight is polarized by scattering and reflection, and many animals take advantage of this to guide tasks such as orientation, prey detection and water surface detection (*Wehner, 2001*). To detect polarization patterns in the sky, many insects use a specialised region in the eye, the Dorsal Rim Area (DRA) (*Labhart & Meyer, 1999*). In flies, the DRA is a narrow band of ommatidia along the dorsal margin of the eye, containing specialised central photoreceptors R7 and R8 (*Wada, 1974a*). The microvilli of rhabdomeric photoreceptors are intrinsically dichroic (*Moody & Parriss, 1961*; *Snyder & Laughlin, 1975*), but elsewhere in the eye

Corresponding author
Simon B. Laughlin, sl104@cam.ac.uk

the polarization sensitivity (PS) of the photoreceptors is suppressed by rhabdomere twist (*Smola & Tscharntke, 1979*). In the DRA, however, the rhabdomeres of R7 and R8 are not twisted (*Wunderer & Smola, 1982a*), and thus they present high PS (*Hardie, 1984*).

In the DRA, R7 and R8 form a pair of orthogonal polarization analysers with identical UV sensitivities, sampling the same small area of the sky (*Hardie, 1985*). The axons of R7 and R8 in the same DRA ommatidium project retinotopically to a specific region within the medulla (*Strausfeld & Wunderer, 1985*; *Fortini & Rubin, 1991*), where their signals are compared to extract information about polarization. This comparison most likely involves polarization-opponent interactions (*Weir et al., 2016*), which are also found in higher order neurons in some species of ants, locusts and crickets (*Labhart, 1988*; *Labhart, 2000*; *Labhart, Petzold & Helbling, 2001*; *Vitzthum, Müller & Homberg, 2002*).

Several conflicting parameters determine the quality of the photoreceptor signal. In most flies, the photosensitive membranes—rhabdomeres—of R7 and R8 form a tiered waveguide, the central rhabdomere pair (CRP, Fig. 1). The pair of rhabdomeres lie at the center of the ommatidium, and in axial section they are surrounded by the R1–6 achromatic rhabdomeres that form a waveguide each. The rhabdomere of R7 sits on top of the rhabdomere of R8, and thereby filters the light available to R8. A longer R7 rhabdomere increases the PS of R8 (*Snyder, 1973*; *Gribakin & Govardovskii, 1975*; *Menzel, 1975*; *Hardie, 1984*) but reduces its own PS by self-screening (*Snyder, 1973*; *Nilsson, Labhart & Meyer, 1987*). The quality of the signal coded by a photoreceptor is also limited by the quantal nature of light. A longer rhabdomere will be less affected by photon noise than a shorter one because it absorbs more photons (*Warrant & Nilsson, 1998*), and has more transduction units (microvilli) to convert photons into electrical signals (*Howard, Blakeslee & Laughlin, 1987*; *Anderson & Laughlin, 2000*). However, when the rhabdomeres of R7 and R8 make a CRP of given length it is not possible to increase the length of both photoreceptors simultaneously, lengthening one shortens the other.

To investigate how these length dependent trade-offs between polarization sensitivity and photon noise influence the ability of the fly DRA to code polarization, we used a modelling procedure adapted from color vision (*Vorobyev et al., 1998*; *Osorio & Vorobyev, 1996*) and recently applied to polarization vision (*How & Marshall, 2014*). We find that for a CRP of fixed length, polarization sensitivity and signal coded by a polarization-opponent unit are highest when R8 is as short as possible. On the other hand, measures of discrimination that consider signal and two forms of noise, photon and intrinsic, are highest for intermediate values of R7 and R8 lengths, broadly agreeing with the experimental evidence (*Wada, 1974a*; *Wunderer & Smola, 1982a*). We conclude that it is essential to consider photon and intrinsic noise when analyzing the ability of tiered photoreceptors to support behaviour. When this is done we see that the distribution of a sensor resource, photoreceptor length, can be optimized for opponent coding.

## METHODS

### Photon absorption rates and polarization sensitivity
An optical model gives the rates at which R7 and R8 absorb photons when sampling a small patch of blue sky.
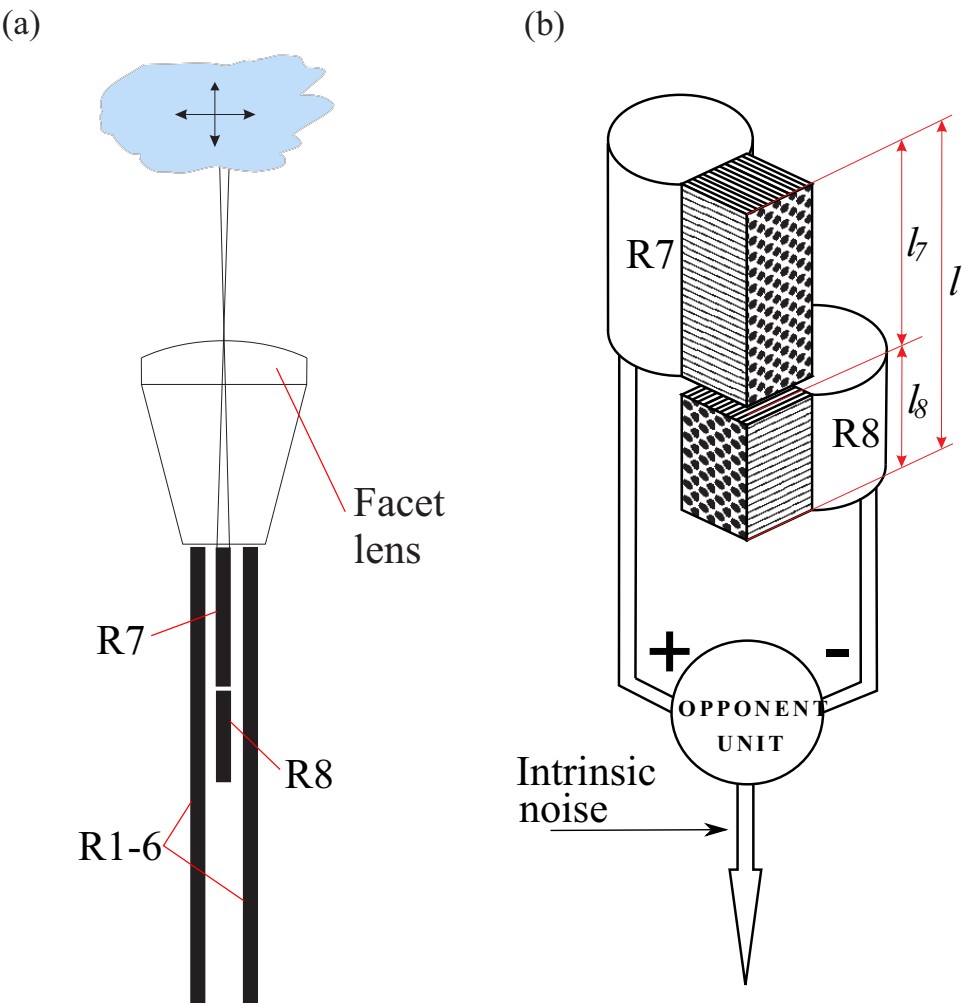

(a)

(b)

R7

Facet lens

R7

R8

R1-6

$l_7$

$l$

$l_8$

R8

**+**   **-**

OPPONENT UNIT

Intrinsic noise

**Figure 1   Optical sampling and opponent coding by R7 and R8 in fly DRA.** (A) Diagramatic longitudinal section of an ommatidium in DRA showing R7/R8's central rhabdomere pair, with two of six outer photoreceptors, type R1–6. Note central rhabdomere pair is much shorter than R1–6 rhabdomeres. R7 and R8 sample the same small area of the sky, focused onto R7 by the facet lens. R7 filters light received by R8. (B) Opponent coding. R8's microvilli are perpendicular to R7's, giving orthogonal polarization sensitivities. Opponent unit outputs difference between inputs from R7 and R8, and adds intrinsic noise.

### Photons available from skylight

To sample a small patch of sky, the facet lens focuses skylight onto the entrance aperture of R7 (*Fig. 1*). The spectral flux of photons at R7's entrance aperture (*Kirschfeld, 1974*; *Land, 1981*) is

$$R_i(\lambda) = \left(\frac{\pi}{4}\right)^2 \left(\frac{1}{F}\right)^2 D_r^2 L(\lambda) \tag{1}$$

where $F$, the facet lens's F-ratio, is the focal length of the facet lens divided by its diameter. $D_r$ is the rhabdomere diameter and $L(\lambda)$ is the spectral radiance of skylight.

The spectral radiance of the sky is equivalent to the radiance of an ideal diffusely reflecting (Lambertian) surface illuminated by cloudless sky, with the sun occluded (*Johnsen, 2012*,
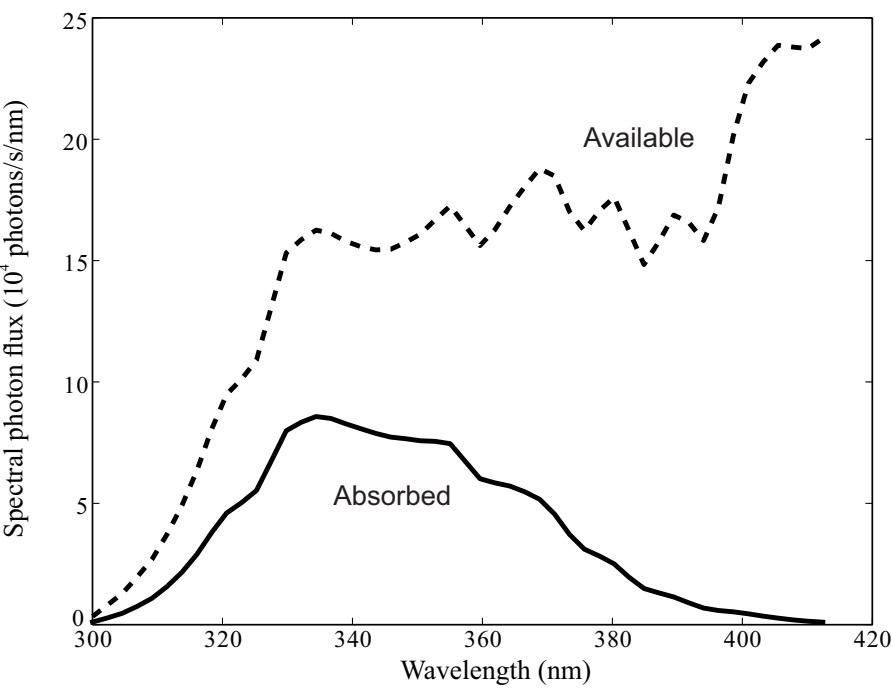

**Figure 2** **Spectral photon flux delivered by facet lens to tip of R7 rhabdomere, $R_i(\lambda)$, (dashed line), and photons absorbed by rhodopsin molecules in R7 and R8 (solid line).** Note absorption is negligible below 300 nm and above 412 nm. Lens views cloudless blue sky which has the spectral irradiance measured at Pretoria, in summer (*Kok, 1972*). Lens F-ratio, $F = 2$; rhabdomere diameter $D_r = 1.55\,\mu$m; central rhabdomere pair length (length of R7 plus R8) $l = 100\,\mu$m. Absorption by rhodopsin with single peak at wavelength 335 nm calculated using an absorption template of rhodopsin (*Stavenga, Smits & Hoenders, 1993*), with a maximum absorption coefficient $k(\lambda_{max}) = 0.0075\,\mu\text{m}^{-1}$.

Chapter 9). To obtain spectral radiance we converted suitable measurements of spectral irradiance (W m$^{-2}$ nm$^{-1}$)—taken at noon on a cloudless summer's day in Pretoria, South Africa (*Kok, 1972*)—to spectral radiance $L(\lambda)$, (photons sr$^{-1}$ m$^{-2}$ nm$^{-1}$) by dividing by $\pi$ and the quantal energy $h\nu = hc/\lambda$.

Consequently the flux of photons at R7's entrance aperture is

$$N_i = \int R_i(\lambda)\,d\lambda = \int \left(\frac{\pi}{4}\right)^2 \left(\frac{1}{F}\right)^2 D_r^2 L(\lambda)\,d\lambda. \tag{2}$$

R7 and R8 capture photons with a UV rhodopsin that only absorbs significantly between 300 nm and 412 nm (Fig. 2). We integrate between these limits. Given measured values for $F$ and $D_r$ (see below, Choice of parameters) and the spectral radiance for skylight (as obtained above), $N_i$ is $1.6 \times 10^7$ photons s$^{-1}$ on a bright summer's day. Note that symbols and their units are listed in Table 1.

### Photon absorption rates

R7's photon absorption rate, $A_7$, depends upon $N_i$, the photon flux incident on R7's entrance aperture, the absorption coefficients of R7's rhabdomere and its length. To account for polarization we decompose the light entering R7 into a pair of orthogonal components, one parallel and the other perpendicular to R7's microvilli. Given an incident

**Table 1** Table of symbols.

| Symbol | Meaning | Units |
|---|---|---|
| $A_s$ | Absorption rate in a small section of the photoreceptor | photons s$^{-1}$ |
| $A_7, A_8$ | Absorption rates in R7 and R8 | photons s$^{-1}$ |
| $D_r$ | Rhabdomere diameter | μm |
| $d$ | Degree of polarization | |
| $\Delta S$ | Number of discriminable polarization angles | |
| $\delta$ | Dichroic ratio of the rhabdomere | |
| $F$ | F-ratio of the facet lens | |
| $F_a(\kappa)$ | Absorptance for wide spectrum light, where $\kappa = l_r k(\lambda_{\max})$ is the product of rhabdomere length and peak absorption coefficient | |
| $I(Q;\theta)$ | Mutual information between opponent unit output and polarization angle | bits |
| $k_\parallel(\lambda)$ | Absorption coefficient for light of wavelength $\lambda$ polarized parallel to the microvilli | μm$^{-1}$ |
| $k_\perp(\lambda)$ | Absorption coefficient for light of wavelength $\lambda$ polarized perpendicular to the microvilli | μm$^{-1}$ |
| $L(\lambda)$ | Spectral radiance | photons sr$^{-1}$ m$^{-2}$ nm$^{-1}$ |
| $l$ | Total length of central rhabdomere pair (CRP) | μm |
| $l_r, l_7, l_8$ | Length of a rhabdomere / R7 rhabdomere / R8 rhabdomere | μm |
| $\lambda$ | Wavelength of light | nm |
| $M_7, M_8$ | Transduction rates | photons s$^{-1}$ |
| $M_7^{(\tau)}, M_8^{(\tau)}$ | Photons transduced per integration time $\tau$ by R7 and R8 | photons |
| $N_i$ | Photon flux incident at the distal extreme of R7 | photons s$^{-1}$ |
| $PS, PS_7, PS_8$ | Polarization sensitivity of a photoreceptor / of R7 / of R8 | |
| $q, q_7, q_8$ | Contrast signal in a photoreceptor / R7 / R8 | |
| $Q$ | Opponent signal | |
| $R_i(\lambda)$ | Spectral photon flux incident at the distal extreme of R7 | photons nm$^{-1}$ |
| $SNR$ | Signal-to-noise ratio | |
| $\sigma_{in}$ | Standard deviation of intrinsic noise | |
| $t_d$ | Photoreceptor dead time | ms |
| $\tau$ | Integration time | ms |
| $\theta$ | Polarization angle | degrees |

flux of $N_i$ photons s$^{-1}$, partially polarized, with degree $d$ and angle $\theta$,

$$N_\parallel = N_i \frac{1 + d\cos(2\theta)}{2} \tag{3}$$

and

$$N_\perp = N_i \frac{1 - d\cos(2\theta)}{2}. \tag{4}$$

We will first consider the case of monochromatic light. If $l_7$ is the length of R7's rhabdomere, and $k_\parallel$ and $k_\perp$ are the absorption coefficients for light polarized parallel and perpendicular to the microvilli, it follows that for light of wavelength $\lambda$ the absorption rate

of R7 is

$$A_7(\lambda) = (1 - e^{-k_\parallel(\lambda)l_7})N_\parallel(\lambda) + (1 - e^{-k_\perp(\lambda)l_7})N_\perp(\lambda). \tag{5}$$

R8 receives the light that passes through R7 with microvilli that are perpendicular to R7's. Consequently, R8's photon absorption rate is given by *Snyder (1973)*:

$$A_8(\lambda) = e^{-k_\parallel(\lambda)l_7}(1 - e^{-k_\perp(\lambda)l_8})N_\parallel(\lambda) + e^{-k_\perp(\lambda)l_7}(1 - e^{-k_\parallel(\lambda)l_8})N_\perp(\lambda). \tag{6}$$

We assume that the rhabdomeres of R7 and R8 have the same absorption coefficients $k_\parallel(\lambda)$ and $k_\perp(\lambda)$, and a dichroic ratio

$$\delta = k_\parallel(\lambda)/k_\perp(\lambda) \tag{7}$$

that is independent of wavelength. Both absorption coefficients peak at a wavelength, $\lambda_{max} = 335$ nm (*Hardie & Kirschfeld, 1983*; *Hardie, 1985*).

### Non-monochromatic light absorption

Skylight is not monochromatic and thus Eqs. (5) and (6) are not valid for skylight absorption. To obtain the absorption of non-monochromatic light by R7 and R8, we must integrate absorption across different wavelengths (*Johnsen 2012*, Chapter 4). Whereas the absorption of monochromatic light is an exponential function of rhabdomere length $l_r$, e.g., $l_7$ in Eq. (5), total absorption is not. The absorptance, the fraction of absorbed photons, increases with rhabomere length, $l_r$, according to a function whose form is uniquely dependent upon the rhabdomere absorption spectrum and the spectrum of incoming light.

$$F_a(\kappa) = \frac{1}{N_i} \int (1 - e^{-k(\lambda)l_r})R_i(\lambda)d\lambda \tag{8}$$

where $N_i$ is the incident photon flux integrated across wavelength (Eq. (2)).

The fraction of photons absorbed by a photoreceptor rhabdomere of length $l_r$ and maximum absorption coefficient $k(\lambda_{max})$ increases with the product of $k(\lambda_{max})$ and $l_r$, $\kappa = k(\lambda_{max})l_r$ according to the function, $F_a(\kappa)$. To simplify our calculations we followed *Warrant & Nilsson (1998)* and approximated $F_a(\kappa)$ with a simple function.

We approximated $F_a(\kappa)$ as follows. Using the absorption template of rhodopsin (*Stavenga, Smits & Hoenders, 1993*) with a single peak absorption at 335 nm (*Hardie & Kirschfeld, 1983*), and $L(\lambda)$, the spectral radiance of blue skylight, we calculated the fraction of absorbed photons for 50 values of $\kappa$. We then fitted a function to these points,

$$F_a(\kappa) = (1 - e^{-\kappa})\left[0.4697838 + 0.05512361\kappa - 0.00291346\kappa^2\right] \tag{9}$$

which approximated the true length dependency with relative error <1%, for photoreceptor lengths up to 1 mm (Fig. 3).

By comparison, adapting the *Warrant & Nilsson (1998)* approximation by extrapolating its coefficients would produce a reasonable approximation for medium length photoreceptors (Fig. 3) but would not have produced the right behaviour in the limit of very short photoreceptors. Consequently, the PS of very short photoreceptors calculated with the *Warrant & Nilsson (1998)* approximation would not be its intrinsic PS, i.e., its dichroic ratio $\delta$.

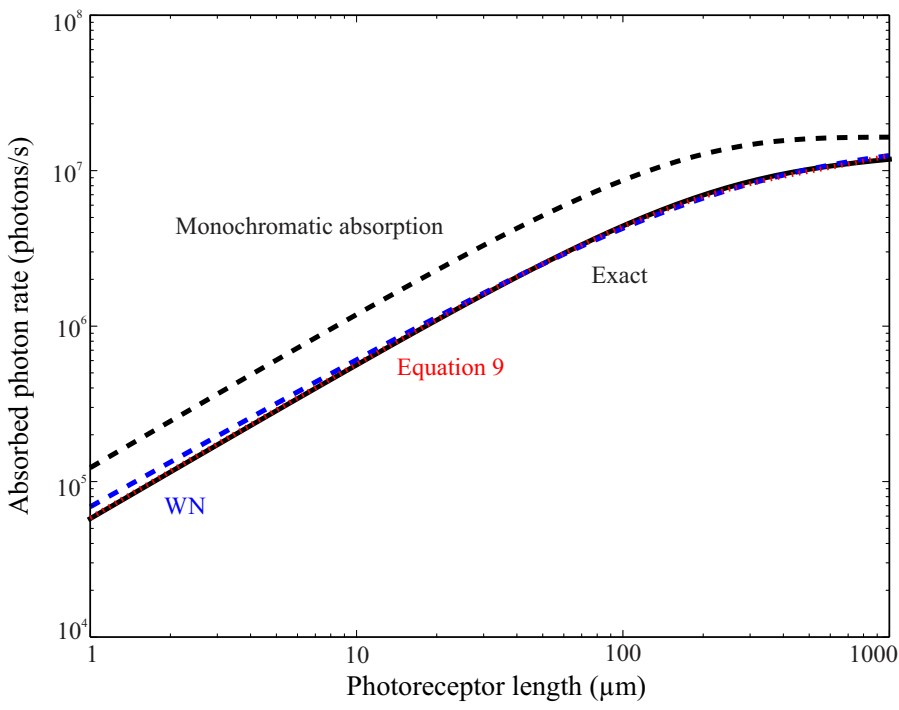

**Figure 3 Absorption of blue skylight by UV rhodopsin, $N_i F_a(\kappa)$, as a function of length $l_r$.** The exact calculation (black solid line) is very different from exponential absorption at the peak wavelength (black dashed line). Our approximation to $F_a(\kappa)$, (Eq. (9), red dotted line), fits better at very short lengths than approximation made by *Warrant & Nilsson (1998)* (blue dashed line). Parameters used for calculations as in subscript to Fig. 2.

### Absorption rates in a tiered central rhabdomere pair receiving polarized skylight

Equations (5) and (6), which give the absorption of polarized monochromatic light by R7 and R8, are valid for all wavelengths. Thus we can use our approximation of absorbtance (Eq. (9)) to calculate the rates at which R7 absorb photons when receiving polarized skylight perpendicular or parallel to the microvilli of R7:

$$A_{\parallel 7} = \int (1 - e^{-k_\parallel(\lambda) l_7}) R_\parallel(\lambda) \mathrm{d}\lambda = F_a(\kappa_{\parallel 7}) N_\parallel \tag{10a}$$

$$A_{\perp 7} = \int (1 - e^{-k_\perp(\lambda) l_7}) R_\perp(\lambda) \mathrm{d}\lambda = F_a(\kappa_{\perp 7}) N_\perp \tag{10b}$$

where we use the shorthand $\kappa_{\parallel 7} = k_\parallel(\lambda_{\max}) l_7$ and $\kappa_{\perp 7} = k_\perp(\lambda_{\max}) l_7$.

For light partially polarized, the R7 absorption rate is the sum of the absorption rate of light polarized parallel ($A_{\parallel 7}$) and perpendicular ($A_{\perp 7}$) to R7 microvilli

$$A_7 = F_a(\kappa_{\parallel 7}) N_\parallel + F_a(\kappa_{\perp 7}) N_\perp \tag{11}$$

where $N_\parallel = \int R_\parallel(\lambda) \mathrm{d}\lambda$ and $N_\perp = \int R_\perp(\lambda) \mathrm{d}\lambda$ are the photon rates incident on the distal tip of R7 with wavelengths between 300 nm and 412 nm, and polarized parallel and perpendicular to the microvilli of R7. $N_\parallel$ and $N_\perp$ depend on the incident photon rate, $N_i$,
the degree of polarization, $d$, and the angle of polarization, $\theta$ (Eqs. (3) and (4)). Thus Eq. (11) tells us how the absorption rate of R7, $A_7$, depends on $N_i$, $d$ and $\theta$.

To calculate the absorption rates of R8, we must take account of the light absorbed by R7. To do this we first calculate the absorption rates of the CRP (R7 plus R8) for light polarized parallel and perpendicular to R7 microvilli:

$$A_{\|7} + A_{\|8} = \int \left[1 - e^{-k_{\|}(\lambda)l_7} e^{-k_{\perp}(\lambda)l_8}\right] R_{\|}(\lambda)\mathrm{d}\lambda = F_a\left(\kappa_{\|7} + \kappa_{\perp8}\right)N_{\|} \tag{12a}$$

$$A_{\perp7} + A_{\perp8} = \int \left[1 - e^{-k_{\perp}(\lambda)l_7} e^{-k_{\|}(\lambda)l_8}\right] R_{\perp}(\lambda)\mathrm{d}\lambda = F_a\left(\kappa_{\perp7} + \kappa_{\|8}\right)N_{\perp} \tag{12b}$$

where we use the shorthand $\kappa_{\|8} = k_{\|}(\lambda_{\max})l_8$ and $\kappa_{\perp8} = k_{\perp}(\lambda_{\max})l_8$.

We then subtract the absorption rates of R7, $A_{\|7}$ or $A_{\perp7}$ to obtain the absorption rates of R8 for light polarized parallel and perpendicular to R7 microvilli, $A_{\|8}$ and $A_{\perp8}$. Summing these two components, $A_{\|8}$ and $A_{\perp8}$, gives the absorption rate of R8:

$$A_8 = A_{\|8} + A_{\perp8} = \left[F_a(\kappa_{\|7} + \kappa_{\perp8}) - F_a(\kappa_{\|7})\right]N_{\|} + \left[F_a(\kappa_{\perp7} + \kappa_{\|8}) - F_a(\kappa_{\perp7})\right]N_{\perp}. \tag{13}$$

Because we know how $N_{\|}$ and $N_{\perp}$ depend on $N_i$, $d$ and $\theta$ (Eqs. (3) and (4)), Equation (13) gives the dependence of the absorption rate of R8, $A_8$, on $N_i$, $d$ and $\theta$.

## Polarization sensitivity, signal and noise
### Transduction and absorption rates
Sensitivity, signal and noise depend upon the transduction rate $M$; the rate at which a photoreceptor transduces absorbed photons to quantum bumps (unitary electrical responses to single photons). At all but the highest light levels, a fly photoreceptor transduces a constant high proportion ($>0.5$) of absorbed photons (*Dubs, Laughlin & Srinivasan, 1981*; *Howard, Blakeslee & Laughlin, 1987*; *Van Steveninck & Laughlin, 1996a*). For simplicity we assume that $M = A$. At the highest light levels transduction units (microvilli) saturate and this reduces sensitivity, reduces signal, and changes the statistics of noise (*Howard, Blakeslee & Laughlin, 1987*; *Song et al., 2012*). To account for these effects we convert absorption rate, $A$, to transduction rate, $M$, using our transduction unit saturation model (see below).

### Polarization sensitivity without transduction unit saturation
In the absence of transduction unit saturation, transduction rates $M$ equal absorption rates $A$. Because the microvilli of R7 and R8 are well aligned, the polarization angles at which transduction rates are maximum and minimum, $\theta_{\max}$ and $\theta_{\min}$, are parallel and perpendicular to the microvilli, respectively. Thus $\theta_{\min} = 90° + \theta_{\max}$.

Polarization sensitivity is the ratio between the maximum and minimum transduction rates produced by linearly polarized light (degree of polarization $d = 1$) of constant intensity (*Snyder, 1973*):

$$PS = \frac{M(\theta_{\max}, d = 1)}{M(\theta_{\min}, d = 1)}. \tag{14}$$

Note that for simplicity we present generic equations in which $M$ stands for either $M_7$ or $M_8$, depending on whether it is calculated using $A_7$ or $A_8$.

### Signal, noise and contrast; unsaturated regime

To calculate signal and noise, we follow opponent models of color coding (*Osorio & Vorobyev, 1996*; *Vorobyev & Osorio, 1998*), and normalise photon transduction rate, $M(\theta)$, to the background rate, $M_{bg}$

$$q(\theta) = \frac{M(\theta)}{M_{bg}}. \tag{15}$$

The background is unpolarized light with the same spectrum and intensity, $N_i$. We take $M_{bg}$ to be the mean transduction rate, in which case $q(\theta)$ is a contrast signal.

Given a constant photon flux $N_i$ with polarization degree $d$, the transduction rate $M(\theta)$, and hence the contrast signal $q(\theta)$, varies with the polarization angle $\theta$ as $\cos(2\theta)$ or, equivalently, $\cos^2(\theta)$.

$$q(\theta) = 1 + d\,\frac{PS-1}{PS+1}\,\cos 2(\theta - \theta_{max}). \tag{16}$$

Note that the background ($d = 0$) produces the same quantum catch as light polarized at $45°$ to the preferred absorption axis ($\theta_{max} = 45°$).

The total range of contrast signals that can be produced by changes in polarization angle, $\Delta q$, depends on the degree of polarization, $d$, and the $PS$, according to Eq. (16):

$$\Delta q = q(\theta_{max}) - q(\theta_{min}) = 2d\frac{PS-1}{PS+1}. \tag{17}$$

The reliability of an optical signal is limited by photon noise, random fluctuations in absorption rate that follow the Poisson distribution. In the absence of transduction unit saturation we assume that every absorbed photon produces a quantum bump. Thus during a time interval $\tau$, a photoreceptor transduces $\mathbf{M^{(\tau)}}(\theta)$ photons, drawn from a Poisson distribution with mean $M^{(\tau)}(\theta) = \tau M(\theta)$. By definition, the noise variance equals the mean:

$$\mathrm{Var}(\mathbf{M^{(\tau)}}(\theta)) = \langle \mathbf{M^{(\tau)}}(\theta) \rangle = \tau M(\theta). \tag{18}$$

Note that for the rest of Methods we write random variables in bold, and denote their mean values with regular letters.

We define the signal to noise ratio, $SNR$, as the ratio between the range of signals produced by changes in polarization angle and the standard deviation of photon noise. This definition holds for both quantum catches $\Delta M^{(\tau)}$ and contrast signals $\Delta q$:

$$SNR = \frac{\Delta M^{(\tau)}}{\sqrt{\mathrm{Var}(\mathbf{M^{(\tau)}}(\theta))}} = \frac{\Delta q}{\sqrt{\mathrm{Var}(\mathbf{q^{(\tau)}}(\theta))}}. \tag{19}$$

Assuming that photon noise is independent of the contrast signal

$$SNR = \frac{\Delta q}{1/\sqrt{\tau M_{bg}}} = 2d\frac{PS-1}{PS+1}\sqrt{\tau M_{bg}}. \tag{20}$$

This assumption is valid for small contrast signals, as produced at low degrees of polarization, $d$.

## Transduction unit saturation, sensitivity, signal and noise

In fly photoreceptors each microvillus acts as a transduction unit, producing an all or nothing unitary response (a quantum bump) to the absorption of a single photon (*Hardie & Raghu, 2001*). To model the effects of transduction unit saturation we divide the rhabdomere into thin (1 µm) segments and use extensions of Eqs. (11) and (13) to obtain the mean and variance of photon absorption in each segment. This treatment accounts for the change in mean absorption rate along the rhabdomere—light entering all but the first section is filtered by the sections above it.

Saturation occurs because it takes time for a microvillus to reload after it produces a quantum bump. This dead time, $t_d$, sets the minimum interval between quantum bumps produced by a microvillus. Thus at high light levels a microvillus's transduction rate fails to keep up with its absorption rate. Consider a 1 µm segment of rhadomere with $n_m$ microvilli, absorbing an average of $A_s$ photons per second. Photons are absorbed in each of the $n_m$ microvilli with Poisson probabilities of parameter $v = A_s t_d / n_m$, but when more than one photon is absorbed in the time interval $t_d$, only one photon is transduced. As a consequence, during the interval of time $t_d$, each microvillus either transduces one photon with probability

$$\mathcal{P}(1) = 1 - e^{-v} \tag{21}$$

or does not transduce it, with a probability

$$\mathcal{P}(0) = e^{-v}. \tag{22}$$

Considering the $n_m$ microvilli in a single segment of rhabdomere, the number of photons transduced $\mathbf{M}^{(\tau)}$ during an integration time $\tau$ (which we take to be an integer multiple of $t_d$) follows a binomial distribution with success probability $1 - e^{-v}$ and number of trials $n_m \tau / t_d$. This binomial has a mean

$$\langle \mathbf{M}^{(\tau)} \rangle = (1 - e^{-v}) n_m \tau / t_d \tag{23}$$

and variance

$$\mathrm{Var}(\mathbf{M}^{(\tau)}) = e^{-v}(1 - e^{-v}) n_m \tau / t_d. \tag{24}$$

Note that at low light levels, $v = A_s t_d / n_m \ll 1$ and our binomial model approximates the Poisson distribution of the absorbed photons, as expected because at low light levels the effects of saturation on signal and noise are negligible.

$$\langle \mathbf{M}^{(\tau)} \rangle = \mathrm{Var}(\mathbf{M}^{(\tau)}) = A_s \tau. \tag{25}$$

Because photons are transduced independently in each microvillus, and hence in every segment, the mean numbers of photons transduced by R7 and R8, $M_7^{(\tau)}$ and $M_8^{(\tau)}$, and their variances, are obtained by summing the means and variances of all segments.

## Polarization opponent model

To assess the effects of signal and noise on the ability to discriminate angles of polarization, we consider the output of a simple opponent mechanism that subtracts the input from R8 from the input from R7 in the same ommatidium. The opponent unit's output $\mathbf{Q}^{(\tau)}$ is the difference between the two photoreceptor contrast signals, $\mathbf{q}_7^{(\tau)}$ and $\mathbf{q}_8^{(\tau)}$, plus a contribution of intrinsic noise. Thus, for each given polarization angle $\theta$, $\mathbf{Q}^{(\tau)}$ is a random variable whose mean is independent of the integration time $\tau$:

$$\langle \mathbf{Q}^{(\tau)} \rangle = M_7^{(\tau)}(\theta)/M_{7,\mathrm{bg}}^{(\tau)} - M_8^{(\tau)}(\theta)/M_{8,\mathrm{bg}}^{(\tau)} \tag{26}$$

$$= M_7(\theta)/M_{7,\mathrm{bg}} - M_8(\theta)/M_{8,\mathrm{bg}} \tag{27}$$

but has noise variance that depends on $\tau$. The variance of $\mathbf{Q}^{(\tau)}$ will be the sum of noise variances incoming from both R7 and R8, and intrinsic noise variance. Thus, in the absence of saturation

$$\mathrm{Var}(\mathbf{Q}^{(\tau)}) = \mathrm{Var}(\mathbf{q}_7^{(\tau)}) + \mathrm{Var}(\mathbf{q}_8^{(\tau)}) + 2(\sigma_{\mathrm{in}}^{(\tau)})^2 \tag{28}$$

$$= \left[ M_7(\theta)/M_{7,\mathrm{bg}}^2 + M_8(\theta)/M_{8,\mathrm{bg}}^2 + 2\sigma_{\mathrm{in}}^2 \right]/\tau \tag{29}$$

where $\sigma_{\mathrm{in}}^2$ and $(\sigma_{\mathrm{in}}^{(\tau)})^2$ are the variances of intrinsic noise (resulting from Gaussian white noise added to the contrast signal $\mathbf{q}_7$ and $\mathbf{q}_8$ of each of the photoreceptors) in a signal integrated across an interval of 1 s and an interval of $\tau$, respectively.

## Number of discriminable polarization angles: a measure of performance which considers noise

Photon noise and intrinsic noise limit how precisely a polarization angle, $\theta$, can be estimated. To quantify the effect of this limitation we take successful models in colour vision (*Vorobyev & Osorio, 1998*) as a basis, and consider the R7/R8 system as equivalent to a dichromatic space. We define a distance $\Delta S$ between the signals generated in the opponent unit by light partially polarized with the same degree of polarization $d$ and incident photon flux $N_{\mathrm{i}}$ but with polarization angles $\theta_1$ and $\theta_2$:

$$\Delta S = \frac{|Q(\theta_1) - Q(\theta_2)|}{\sqrt{\mathrm{Var}(\mathbf{q}^{(\tau)})}} = \frac{\left| [q_7(\theta_1) - q_7(\theta_2)] - [q_8(\theta_1) - q_8(\theta_2)] \right|}{\sqrt{\mathrm{Var}(\mathbf{q}_{7^{(\tau)}}) + \mathrm{Var}(\mathbf{q}_{8^{(\tau)}}) + 2(\sigma_{\mathrm{in}}^{(\tau)})^2}} \tag{30}$$

where $|\ |$ is the absolute value.

This distance depends both on the amount the opponent signal $Q$ changes for different polarization angles —ultimately determined by the PS of R7 and R8 (Eq. (17))— and the reliability of the opponent signal, i.e., the amount of noise (composed of intrinsic and photon noise). A change in polarization angle $\Delta\theta$ can be detected in the opponent signal when the distance $\Delta S$ is bigger than a given value. For simplicity, we will take this threshold to be 1, so $\Delta S$, as defined in Eq. (30), is directly the number of discriminable polarization angles between polarization angles $\theta_1$ and $\theta_2$.

When considering the number of discriminable polarization angles between angles that produce very different quantum catches, we need to take into account that the
photoreceptor noise varies as one moves across the stimulus space. The stimulus space is thus endowed of a Riemannian metric (*Wyszecki & Stiles, 1982*). The distance between light polarized at 0 degrees and light polarized at 90 degrees following a path of constant degree of polarization is better approximated by *Osorio & Vorobyev (1996)*:

$$\Delta S = \sum_{i=0}^{n-1} \frac{\left| [q_7(\theta_i) - q_7(\theta_{i+1})] - [q_8(\theta_i) - q_8(\theta_{i+1})] \right|}{\sqrt{\mathrm{Var}(\mathbf{q}_7^{(\tau)}(\theta_i)) + \mathrm{Var}(\mathbf{q}_8^{(\tau)}(\theta_i)) + 2(\sigma_{\mathrm{in}}^{(\tau)})^2}} \tag{31}$$

with $\theta_0 = 0 < \theta_1 < \cdots < \theta_{n-1} < \theta_n = 90$ degrees.

## Mutual information: a different measure of performance

We use information theory to define a second measure of the ability to discriminate between different polarization angles. This new measure quantifies how much we can reduce our uncertainty on the polarization angle by a single measure of the noisy opponent signal. In principle, it should be related to, but does not necessarily correlate with, measures based on ideal observer performance, such as the total number of discriminable angles (*Thomson & Kristan, 2005*).

Let $\theta \in [0, \pi/2]$ be the polarization angle, $\mathbf{Q}$ the output of a polarization-opponent unit, and $f(\theta), f(Q)$ and $f(Q, \theta)$ their marginal and joint probability density functions. We assume that the distribution of polarization angles $\boldsymbol{\theta}$ is uniform, i.e., $f(\theta) = 2/\pi$, and that the degree of polarization $d$ is constant.

If there was no noise, $\mathbf{Q}$ would be simply a function of $\theta$, $Q(\theta)$. Since the system is limited by noise, we consider the probability density of $\mathbf{Q}$ when we know $\boldsymbol{\theta}$. Under reasonable assumption it is a Gaussian of standard deviation $\sigma_Q = \sqrt{\mathrm{Var}(\mathbf{Q})}$, in our case a function of the angle of polarization $\theta$:

$$f(Q|\theta) = \frac{1}{\sigma_Q(\theta)\sqrt{2\pi}} e^{\frac{-(Q - Q(\theta))^2}{2\sigma_Q^2(\theta)}} \tag{32}$$

$$\sigma_Q^2(\theta) = (M_7(\theta)/M_{7,\mathrm{bg}}^2 + M_8(\theta)/M_{8,\mathrm{bg}}^2 + 2\sigma_{\mathrm{in}})/\tau. \tag{33}$$

The mutual information or rate of transmission of information between the two continuous random variables $\boldsymbol{\theta}$ and $\mathbf{Q}$ is then defined as *Shannon (1948)*

$$I(\mathbf{Q}; \boldsymbol{\theta}) = \int \int f(Q, \theta) \log \frac{f(Q, \theta)}{f(\theta)f(Q)} d\theta dQ \tag{34}$$

which quantifies how much a measure of the opponent output, $\mathbf{Q}$, reduces the uncertainty about the polarization angle, $\boldsymbol{\theta}$, assuming that we are certain about the degree of polarization, $d$.

## Choice of parameters

F-ratio was not measured in the DRA, so we chose $F = 2$, an average value in other parts of *C. vicina* eye (*Hardie, 1985*). We used $D_r = 1.55 \, \mu\mathrm{m}$ (*Wunderer & Smola, 1982a*). Measurements suggest that the absorption coefficient of a photoreceptor rhabdomere for unpolarized light ($\frac{k_{\parallel} + k_{\perp}}{2}$) lies between 0.01 $\mu\mathrm{m}^{-1}$ (e.g., *Hardie, 1984*) and 0.005 $\mu\mathrm{m}^{-1}$

(*Warrant & Nilsson, 1998*), so we took $k = 0.0075 \ \mu m^{-1}$. Measurements of R7/R8 photoreceptors' PS in the DRA of *C. vicina* range from 6 to 19 (*Hardie, 1984*), so we chose a dichroic ratio of $\delta = k_\parallel / k_\perp = 10$. Sky has a maximum polarization degree of about $d = 0.6 - 0.8$ in the UV, at 90 degrees from the sun in clear skies, but smaller at other orientations and under different meteorological conditions (*Barta & Horváth, 2004*). Behavioural threshold was measured to be at a degree of polarization of $d = 0.05$ in crickets and $d = 0.1$ in honeybees (*Barta & Horváth, 2004*). We modelled a polarization degree of $d = 0.1$.

Blowfly R1–6 photoreceptors have around $9 \times 10^4$ microvilli along an average length of 250 μm (*Hardie, 1985*; *Hochstrate & Hamdorf, 1990*). We assumed here a similar linear density of microvilli in the R7 and R8 of the DRA. We chose $t_d = 30$ ms for the minimum interval between transduced photons in a *C. vicina* microvilli (*Hochstrate & Hamdorf, 1990*; *Song et al., 2012*), and we assumed that the fly integrates the signal across three of those intervals, i.e., integration time $\tau = 90$ ms.

## RESULTS

### The division of the central rhabdomere pair between R7 and R8

In the fly dorsal rim (DRA), the two polarization coding photoreceptors, R7 and R8, construct a central rhabdomere pair (CRP), whose length, $l$, is of the order of 100 μm. R7 and R8 divide the CRP between them; R7 constructs the upper part, of length $l_7$ and R8 the lower of length $l_8$ (Fig. 1). To see how the division of the CRP between R7 and R8 determines their ability to code polarization we model three determinants of signal quality, polarization sensitivity, polarization signal and signal to noise ratio, as a function of their length fractions,

$$\hat{l_7} = l_7 / l \tag{35}$$
$$\hat{l_8} = l_8 / l \tag{36}$$

where $l = 100 \ \mu m$.

### The flux of incident photons

Because photoreceptor signal and noise depend upon the numbers of photons absorbed and transduced, we start by establishing the numbers of photons available for absorption. Our optical model calculates the rate at which the facet lens delivers photons from a clear blue sky to the entrance aperture of R7's rhabdomere (Fig. 1; Eqs. (1) and (2)), by taking into account the intensity and wavelength spectrum of skylight, and light gathering by the facet lens and rhabdomere. The spectral photon flux at R7's entrance aperture is integrated between 300 nm and 412 nm to give our measure of incident photon flux $N_i$. Beyond these wavelength limits R7 and R8's UV rhodopsin—peak wavelength 335 nm (*Hardie, 1984*; *Stavenga, Smits & Hoenders, 1993*)—absorbs negligible numbers of photons (Fig. 2).

We use this incident photon flux, $N_i$, as our measure of the intensity of incident light. At noon on a bright summer's day, when the light intensity is approximately $10^5$ lux, $N_i = 1.6 \times 10^7$ photons s$^{-1}$ (Methods 2.1). The lowest intensity we model, $N_i = 100$ photons s$^{-1}$, corresponds to late nautical twilight (*Johnsen, 2012*).

## Polarization sensitivity and the length fractions of R7 and R8

When a photoreceptor is sensitive to the plane of linearly polarized light, absorption varies with the polarization angle, $\theta$. Polarization sensitivity is defined as the ratio between the maximum absorption, at polarization angle $\theta_{max}$ and the minimum absorption at $\theta_{min}$, when illuminated with a constant and completely polarized light. In a rhabdomere that does not twist (as in the DRA), $\theta_{max}$ is parallel to the rhabdomere's microvilli, and $\theta_{min}$ perpendicular.

We use our optical model and our model of transduction unit saturation to show how the PS's of R7 and R8 depend on length fraction. The optical model first calculates the rates at which R7 and R8 absorb photons, $A_7$ and $A_8$, given a flux of incident photons $N_i$, polarized at angle $\theta$ with a degree of polarization $d = 1$ (Eqs. (3), (4), (11) and (13)). This calculation takes into account the known absorption properties of R7 and R8, their lengths and hence length fractions, and the filtering by R7 of the light delivered to R8. Although the PS is usually taken to be the ratio between maximum and minimum absorption rates, $A(\theta_{max})/A(\theta_{min})$ (e.g., *Snyder, 1973*), we go one step further.

We convert absorption rate, $A$, to a measure that is more closely related to a photoreceptor's ability to code information, the photon transduction rate, $M$. For a fly photoreceptor $M$ is the rate at which it generates elementary electrical responses to single photons—quantum bumps. Below $M = 10^4$ photons s$^{-1}$ the quantum efficiency of fly phototransduction is high, $> 0.5$, and constant (Methods; *Dubs, Laughlin & Srinivasan, 1981*; *Van Steveninck & Laughlin, 1996b*). In this case we assume that the quantum efficiency of transduction $= 1$, therefore $M = A$. At higher light levels the transduction units that generate quantum bumps (individual microvilli) saturate. Saturation lowers the quantum efficiency of transduction so that $M$ is significantly less than $A$, reduces signal amplitude and changes the statistics of noise (*Howard, Blakeslee & Laughlin, 1987*; *Song et al., 2012*). We account for these effects with our transduction unit saturation model. By comparing results obtained with and without saturation we establish the low intensity regime in which saturation is negligible, and demonstrate the effects of saturation at higher light levels.

Our model confirms that the polarization sensitivities of R7 and R8, $PS_7$ and $PS_8$, are strongly dependent on the division of resources between R7 and R8 (*Snyder, 1973*). As the length fraction $\hat{l}_8$ decreases, $PS_8$ increases because a longer R7 is a more effective polarization filter, and $PS_7$ decreases because the effects of self-screening increase with length. Thus for a CRP of length $l = 100\ \mu$m, $PS_7$ drops from its dichroic ratio, $\delta = 10$, to 7 and $PS_8$ increases from 7 to 24 (Fig. 4).

$PS_7$ and $PS_8$ are also sensitive to the spectrum of incident light. When illuminated with monochromatic light at peak absorption wavelength, 335 nm, R7 absorbs a higher fraction of incident photons. Consequently $PS_7$ is reduced more by self-screening, and $PS_8$ is increased more by filtering by R7. For example, when $l = 100\ \mu$m and $\hat{l}_8 = 0.5$, $PS_7 = 7.5$ and $PS_8 = 13.9$. When $\hat{l}_8$ is vanishingly small, $PS_8 = 34$, more than three times the dichroic ratio. However, although $PS_8$ changes with wavelength, for any given wavelength or combination of wavelengths, $PS_8$ is always maximum when $\hat{l}_8$ is vanishingly small. It follows that to maximise PS almost all of the CRP should be allocated to R7.

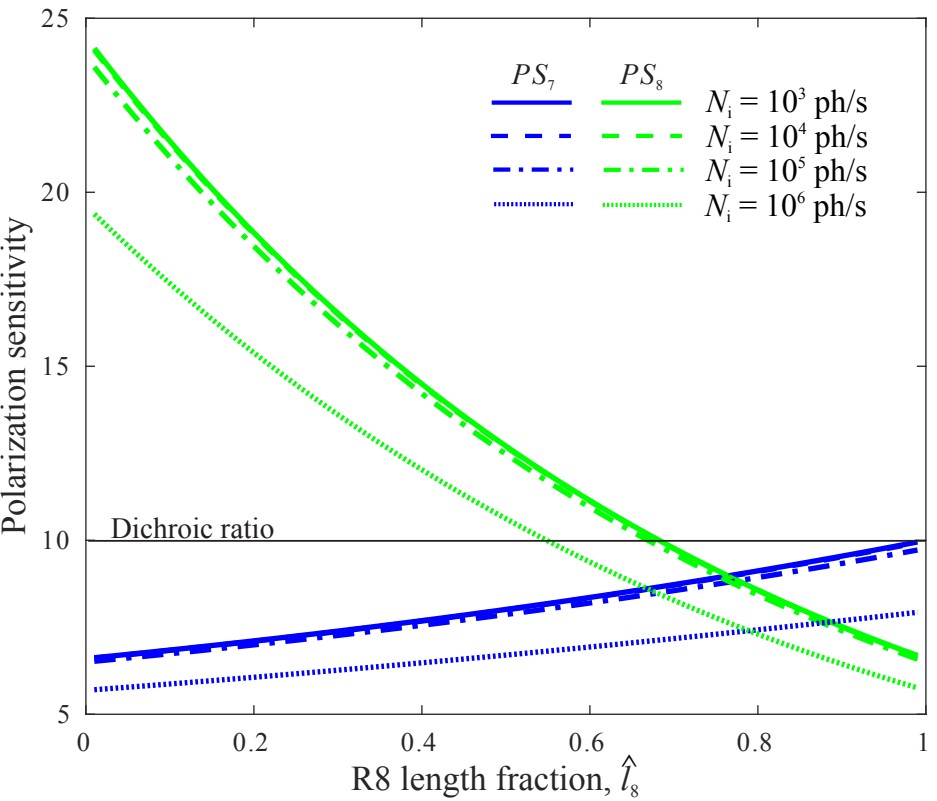

**Figure 4  Polarization sensitivities of R7 and R8, $PS_7$ and $PS_8$, depend on division of CRP between R7 and R8, as specified by R8's length fraction $\hat{l}_8$.** $PS_7$ and $PS_8$ do not vary with light level below $N_i = 10^4$ photons s$^{-1}$. Above this incident photon flux, saturation reduces $PS$.

Transduction unit saturation is significant above $N_i = 10^4$ photons s$^{-1}$, depressing $PS_7$ and $PS_8$ at all length fractions. This result suggests that saturation should be avoided by using the fly's longitudinal pupil (a dense array of small pigment granules that are drawn close to the rhabdomere in bright light) to attenuate the rhabdomeric photon flux (*Anderson & Laughlin, 2000*).

## The dependence of polarization signal amplitude on the length fractions of R7 and R8

Signal amplitude depends upon $M(\theta)$, the relationship between transduction rate and polarization angle. To derive a signal that depends on polarization that is independent of background intensity we follow studies of colour coding (e.g., *Vorobyev & Osorio, 1998*). $M(\theta)$ is normalized by dividing by the mean transduction rate $M_{bg}$ to generate a contrast signal, $q(\theta)$. In the absence of saturation, $q(\theta)$ follows $\cos(2\theta)$, with an amplitude that increases linearly with the degree of polarization $d$ and sub-linearly with $PS$ (Methods; Eq. (16));

$$q(\theta) = 1 + d\,\frac{PS-1}{PS+1}\,\cos 2(\theta - \theta_{max}).$$

The range of contrast signals produced over all polarization angles is given by (Methods; Eq. (17)):

$$\Delta q = q(\theta_{\max}) - q(\theta_{\min}) = 2d\frac{PS-1}{PS+1}.$$

Note that both $q(\theta)$ and $\Delta q$ increase linearly with the degree of polarization, $d$, and, due to the definition of $PS$, sub-linearly with $PS$. This has two consequences. First, $q(\theta)$ only specifies $\theta$ when the degree of polarization, $d$ is constant (*Bernard & Wehner, 1977*; *How & Marshall, 2014*). In the treatment that follows we calculate signals produced when $d = 0.1$. Second, the high values of $PS_8$ that are produced by having a short R8 and long R7 have relatively little effect on $q_8$ and $\Delta q_8$. Indeed, as we will now see, a short R8 is disadvantageous because it suffers badly from photon noise.

## Photon noise, signal to noise ratio and the length fractions of R7 and R8

Photon noise, an inevitable consequence of photon absorption, limits the resolution of photoreceptor signals. To calculate photon noise we integrate the photon transduction rate, $M(\theta)$, over an integration time $\tau$ to obtain a quantum catch $M^{(\tau)}(\theta)$. Because photon noise is Poisson, its variance equals the mean $\tau M(\theta)$. The effect of noise on the resolution of signal depends on the signal to noise ratio, $SNR$. Taking as signal $\Delta q$, we obtain (Methods; Eq. (20))

$$SNR = \frac{\Delta q}{1/\sqrt{\tau M_{\mathrm{bg}}}} = 2d\frac{PS-1}{PS+1}\sqrt{\tau M_{\mathrm{bg}}}.$$

We note (Methods) that the $SNR$ calculated using contrast equals the $SNR$ calculated using transduction rate because, to convert to contrast, both signal and noise are divided by the same factor, $M_{\mathrm{bg}}$.

Like $\Delta q$, $SNR$ increases linearly with $d$ and sub-linearly with the polarization sensitivity $PS$. Also, as in many optical systems limited by photon noise, $SNR$ increases as the square root of mean quantum catch. Consequently $SNR$ is sensitive to both background intensity and photoreceptor length.

Because of this length dependence, $SNR_7$ and $SNR_8$ change greatly with the division of the CRP. Increasing R7's length fraction $\hat{l}_7$ (i.e., decreasing $\hat{l}_8$ in Fig. 5) increases $SNR_7$ as more photons are caught. Thus, a reduction of the effects of photon noise more than compensates for the loss of $PS_7$, and hence signal $\Delta q_7$, across the entire length range. If R7 were to be extended beyond the limit imposed by the length of the DRA's CRP, self-screening would come to dominate. Thus $SNR_7$ would peak and then decline. With a dichroic ratio $\delta = 10$ and a maximum absorption coefficient $k = 0.0075\ \mu\mathrm{m}^{-1}$, as indicated by measurements made on fly photoreceptors, this optimum $SNR_7$ would occur when $l_7 = 200\ \mu\mathrm{m}$. Increasing R8's length fraction has a similar effect; $SNR_8$ increases with $\hat{l}_8$, although with smaller slope. At most length fractions, filtering by R7 increases $SNR_8$ by a small amount. For example, when the CRP is equally divided, $\hat{l}_7 = \hat{l}_8 = 0.5$, $SNR_8$ is slightly greater than $SNR_7$.

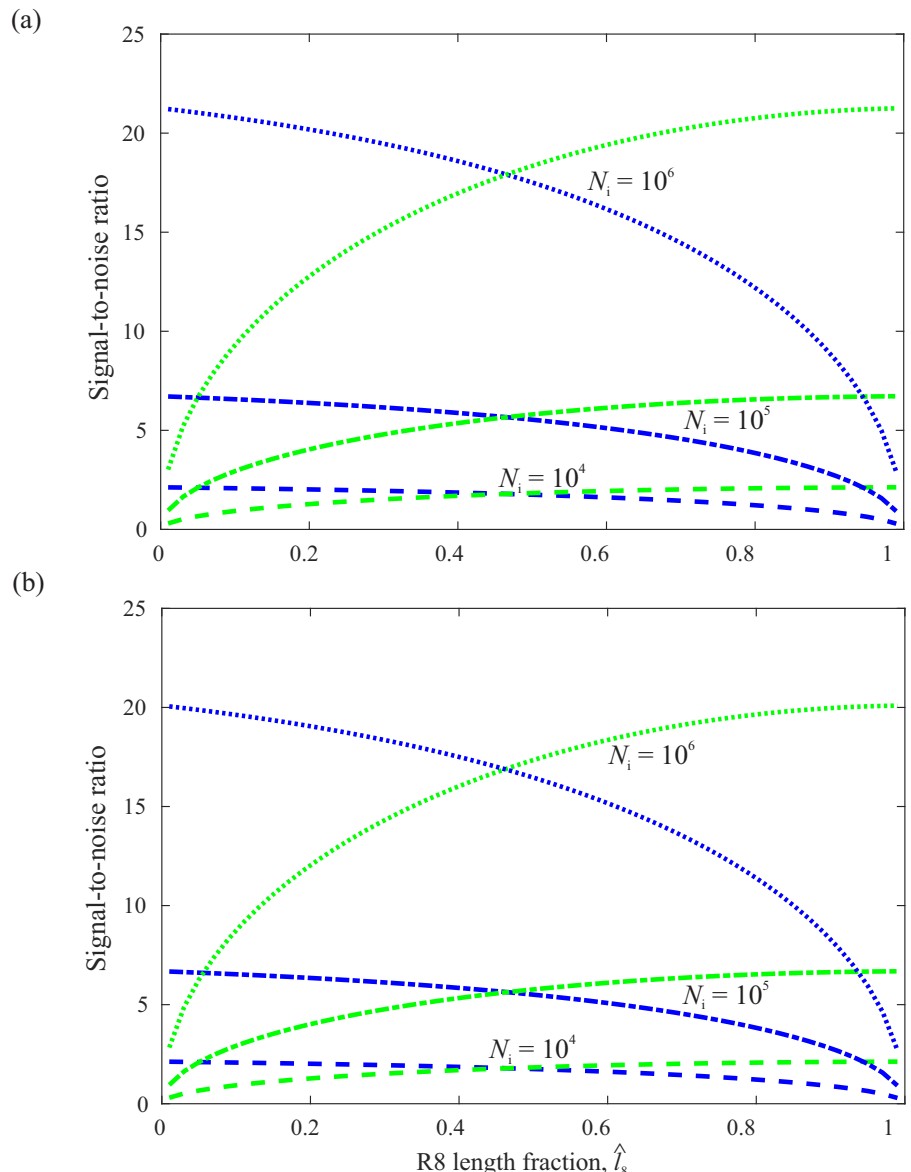

**Figure 5** SNR in R7 (blue) and R8 (green) as a function of the fraction of the CRP occupied by R8, $\hat{l}_8$, calculated for a CRP of fixed length $l = 100\ \mu$m at three light levels, $N_i$. (A) Calculated without modelling the saturation of transduction units. (B) Calculated with saturation. Saturation reduces SNR at the highest intensity. Degree of polarization $d = 0.1$.

Given the dramatic effect of R7 on $PS_8$, its small influence on $SNR_8$ is somewhat surprising, but it is easily explained. Reducing $\hat{l}_8$ reduces R8's quantum catch in two ways. First a shorter R8 absorbs a smaller proportion of the photons delivered by R7. Second, a longer R7 delivers fewer photons. Thus photon catch trumps screening and the division of CRP that maximises both $SNR_7$ and $SNR_8$ is close to equal (Fig. 5). But to what extent would such a division improve the ability of R7 and R8 to code stimuli that are differently polarized? To address this question we use a model of opponent coding.

## An opponent coding model demonstrates optimum length fractions

In our model an opponent unit subtracts the R8 signal from the R7 signal to produce an output signal $Q$. As in color opponent models (e.g., *Osorio & Vorobyev, 1996*), the photoreceptor inputs were independently normalized by dividing by the mean; they correspond to contrast (Eq. (15)). The two previous models of polarization opponency convert light intensity to contrast by taking the logarithm, as do photoreceptors over most of their response range (*Nilsson, Labhart & Meyer, 1987*; *How & Marshall, 2014*).

The opponent unit's output signal is the unweighted difference:

$$Q(\theta) = q_7(\theta) - q_8(\theta) \tag{37}$$

where $q_7(\theta)$ and $q_8(\theta)$ are the contrast signals produced by R7 and R8 when they sample the same small patch of blue sky partially polarized at angle $\theta$.

Just as we defined signal ranges for R7 and R8, the opponent signal range, $\Delta Q$, is given by

$$\Delta Q = \max_{\theta}[Q] - \min_{\theta}[Q] = \Delta q_7 + \Delta q_8. \tag{38}$$

Recall that $\Delta q_7$ and $\Delta q_8$ depend on $PS_7$ and $PS_8$ (Eq. (17)) which in turn depend upon the length fraction $\hat{l}_8$ (Fig. 4). It follows that $\Delta Q$ also varies with length fraction (Fig. 6).

$\Delta Q$ is largest for a vanishingly small length fraction, $\hat{l}_8 \ll 1$ (Fig. 6), mainly because $PS_8$ is maximum. However, although $PS_8$ changes four-fold with $\hat{l}_8$, (Fig. 4) $\Delta Q$ changes by $<7\%$ (Fig. 6). This is because the photoreceptor signal range $\Delta q$ depends on PS via the factor $(PS - 1)/(PS + 1)$ (Eq. (17)), which grows very slowly with PS at the higher values measured in photoreceptors (*Hardie, 1985*). Without saturation, $\Delta Q$ is independent of the incident photon flux $N_i$. Saturation takes effect at intensities in excess of $N_i = 10^5$ photons s$^{-1}$ and progressively reduces $\Delta Q$ by as much as 30%. Saturation also increases the optimum length fraction $\hat{l}_8$, although the optimum is very broad. In summary, the shortest possible R8 generally produces the largest opponent signal because it has the highest PS. However, this configuration is useless. The shortest R8 transduces so few photons that its signal is all but obliterated by photon noise. We must now consider how noise degrades signal in the opponent unit.

## Photon and intrinsic noise limit the resolution of opponent signals

Opponent models of visual coding readily take account of two sources of noise, the photon noise produced when photoreceptors transduce photons and the intrinsic noise produced by subsequent neural mechanisms such as ion channels and synaptic vesicle release (*Faisal, Selen & Wolpert, 2008*). Indeed, intrinsic noise is routinely represented in models of sensory discrimination for two reasons. First, even when presented with noise free inputs a brain cannot discriminate between infinitely many stimuli. Second, intrinsic noise accounts for absolute thresholds.

In our opponent model we express intrinsic noise in terms of equivalent contrast, i.e., a random fluctuation in photoreceptor contrast signal that replicates the effects of intrinsic noise at the opponent unit output. This contrast-referred intrinsic noise is assumed to

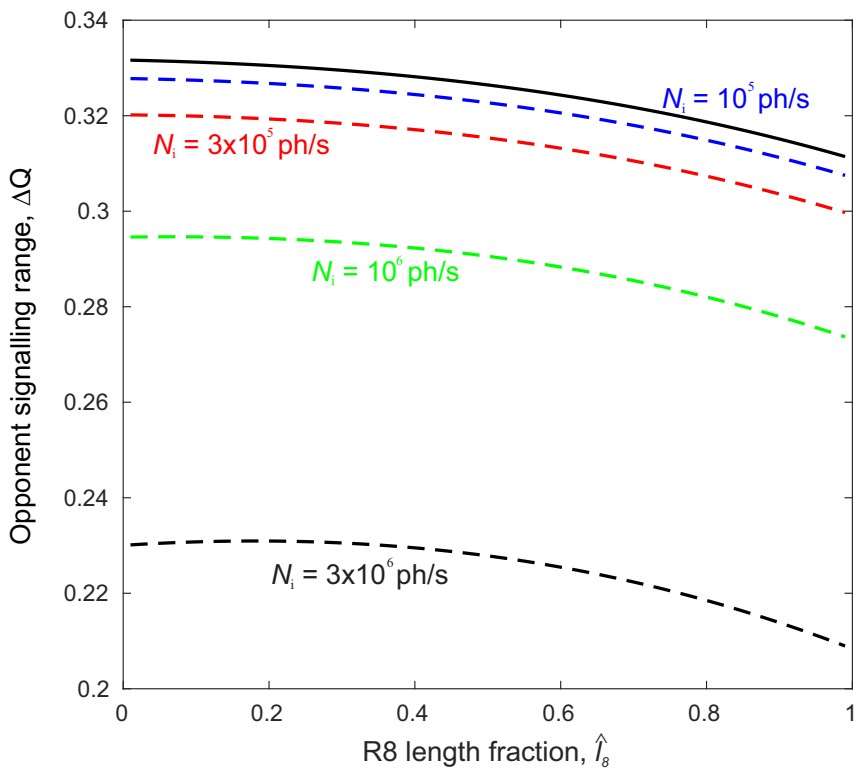

**Figure 6** Polarization-opponent unit's signal range, $\Delta Q$, is largest when the R8 length fraction, $\hat{l}_8$, is vanishingly small and, without transduction unit saturation (solid curve) does not depend on incident photon flux, $N_i$. Saturation progressively decreases $\Delta Q$ with increasing $N_i$ (dashed curves). Rhabdom length, $l = 100 \, \mu m$. Degree of polarization $d = 0.1$. Incident photon flux, $N_i$: $1 \times 10^5$ (blue), $3 \times 10^5$ (red), $1 \times 10^6$ (green) and $3 \times 10^6$ (black) photons $s^{-1}$.

have a Gaussian distribution with zero mean. We set its variance so that intrinsic noise dominates at high light levels and photon noise dominates at lower light levels, as observed in neurons post-synaptic to blowfly photoreceptors, elsewhere in the eye (*Laughlin, Howard & Blakeslee, 1987*; Methods). Photon noise has already been calculated (Methods, Eq. (24)). Without saturation it is Poisson and with saturation it is binomial (*Howard, Blakeslee & Laughlin, 1987*).

The opponent unit combines noise from independent sources, photon noise from R7, photon noise from R8, and intrinsic noise. Consequently noise variances add, even though signals subtract, giving a total noise variance in the opponent output (Methods; Eq. (28))

$$\mathrm{Var}(\mathbf{Q}^{(\tau)}) = \mathrm{Var}(\mathbf{q}_7^{(\tau)}) + \mathrm{Var}(\mathbf{q}_8^{(\tau)}) + 2(\sigma_{in}^{(\tau)})^2$$
$$= \left[ M_7(\theta)/M_{7,bg}^2 + M_8(\theta)/M_{8,bg}^2 + 2\sigma_{in}^2 \right]/\tau.$$

To see how photon and intrinsic noise determine the accuracy with which polarization angle can be coded we calculate the just noticeable difference (jnd) in polarization angle, $\delta\theta$, as a function of polarization angle (*How & Marshall, 2014*). For simplicity, the difference $\delta\theta$ is taken to be "just noticeable" when it produces a change in opponent signal that equals

the standard deviation of the total noise, photon plus intrinsic. Plotting the inverse of $\delta\theta$ gives the discriminability of polarization angle, as a function of $\theta$.

Discriminability is greatest around an angle of $\theta = 45°$ (Fig. 7). Here both R7 and R8 have their highest sensitivity to changes in $\theta$ because $PS$ follows a $\cos^2$ function. Increasing the incident photon flux $N_i$, improves discriminability around all polarization angles, but does not shift the angle of maximum discriminability (Figs. 7A and 7B). Decreasing the length fraction of R8, $\hat{l}_8$, from 0.5 to 0.1 reduces discriminability across all angles. It does so at all three background intensities, but the relative decrease in discriminability is higher at the lower light levels. This observation confirms the importance of photon noise in a short R8. Although shortening R8 increases $PS_8$, and hence its constrast signal $q_8$, the loss of quantum catch reduces $SNR_8$ to such an extent that its reduces the reliability of the opponent output (Figs. 7A and 7B). Transduction unit saturation reduces discriminability at the highest light levels (Fig. 7B).

## Coding ability depends on the length fractions of R7 and R8

The output of the opponent unit, $Q$, gives us two more measures of R7 and R8's ability to code the polarization of the small patch of skylight that is projected onto the tip of the CRP. The first measure is the number of discriminable polarization angles coded by the opponent unit (Eq. (31)). The second measure is the mutual information between polarization angle and opponent signal (Eq. (34)). Our models demonstrate that both measures of coding ability depend upon the division of a resource, a CRP of length $l$, between R7 and R8. As above, the allocation of this resource is specified by the length fraction $\hat{l}_8$.

### Number of discriminable polarization angles varies with the length fractions of R7 and R8

We follow previous studies of the opponent coding of color (*Osorio & Vorobyev, 1996*; *Vorobyev & Osorio, 1998*) and polarization (*How & Marshall, 2014*) and calculate the total number of just noticeable differences in opponent unit output, jnd's, across all polarization angles. This calculation takes into account the fact that a photoreceptor's quantum catch, and hence photon noise, changes with polarization angle (Methods). An opponent signal generated by a pair of photoreceptors with orthogonal $PS$ confounds polarization angle $\theta$ with degree of polarization, $d$ (*Bernard & Wehner, 1977*; *How & Marshall, 2014*). For simplicity and directness we assume $d = 0.1$, thereby assigning all changes in opponent signal, $\Delta Q$, to changes in $\theta$.

The length fraction $\hat{l}_8$ that maximizes the number of discriminable polarization angles depends upon the amplitudes of intrinsic noise and photon noise, and upon transduction unit saturation (Fig. 8). In the absence of intrinsic noise (Fig. 8A), i.e., only photon noise, the number of discriminable polarization angles is maximum when R7 and R8 have approximately equal lengths ($\hat{l}_8 \approx 0.5$), at all light levels. This near equal division is optimal because $SNR$ increases with photoreceptor length (Fig. 5). There is a slight bias towards a longer R7 because this increases $PS_8$. Transduction unit saturation does not change this optimum (Fig. 8A).

The presence of intrinsic noise has no effect on the optimum $\hat{l}_8$ at low and intermediate light levels (Fig. 8B); $N_i = 1 \times 10^5$) and has little effect on the number of discriminable

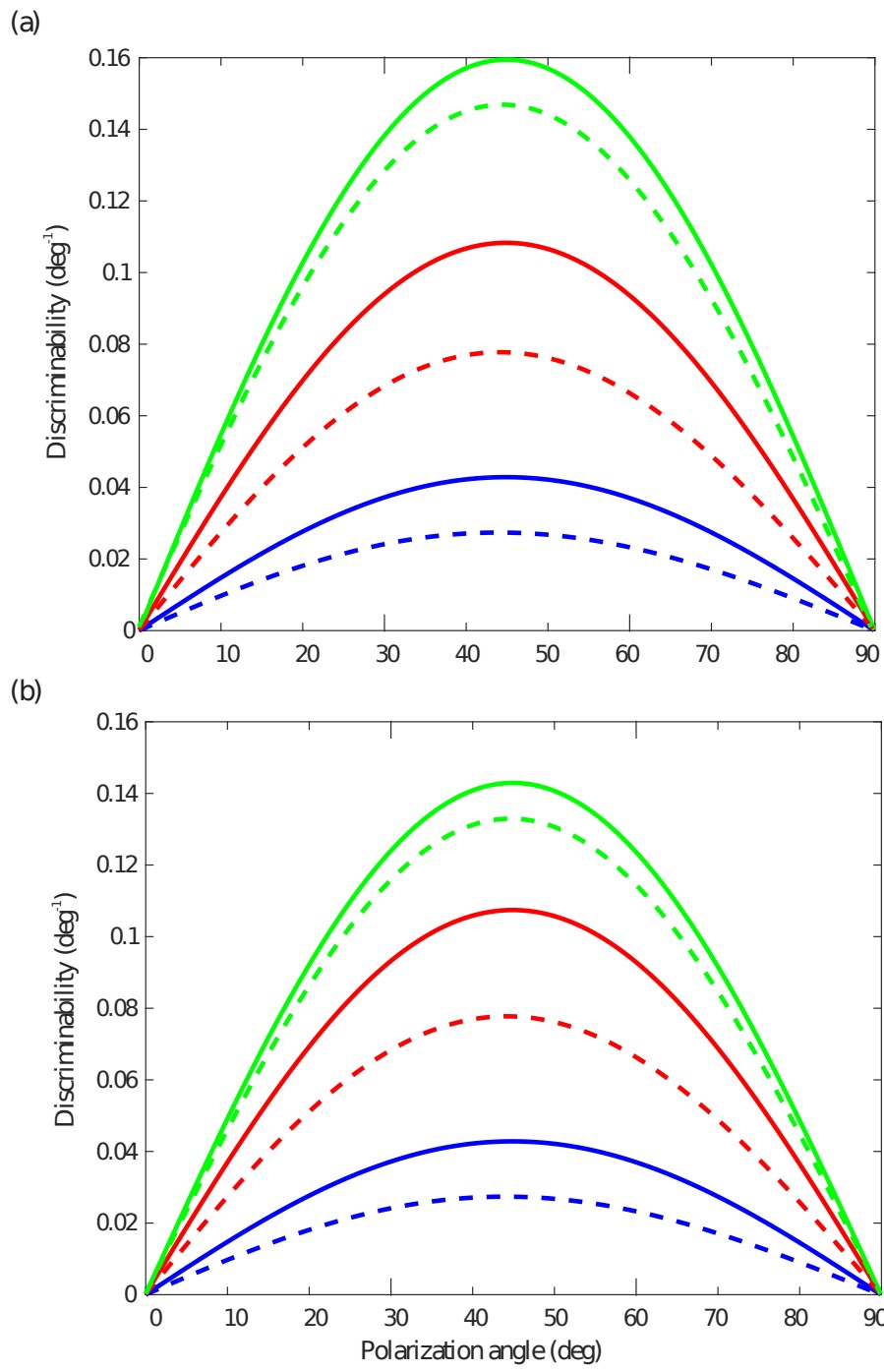

**Figure 7** **Discriminability as a function of polarization angle, $\theta$, for light with degree of polarization $d = 0.1$, at three incident photon fluxes, $N_i$.** (A) Rhabdom length $l = 100\ \mu m$ with R7 and R8 of the same length, $\hat{l}_8 = 0.5$ (continuous lines) and with shorter R8, $\hat{l}_8 = 0.1$ (dashed lines). (B) As (a), but taking into account tranduction unit saturation. Incident photon flux, $N_i$: $1 \times 10^5$ (blue), $3 \times 10^5$ (red), and $1 \times 10^6$ (green) photons $s^{-1}$.

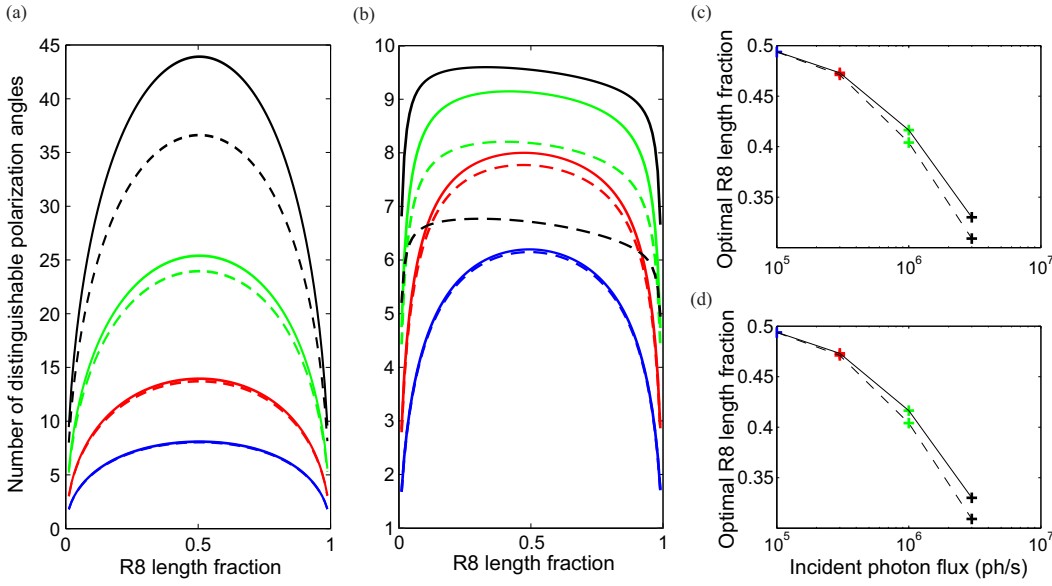

**Figure 8** **Optimum division of CRP between R7 and R8 maximizes two measures of coding ability, number of discriminable polarization angles and mutual information.** Optima depend on light intensity (incident photon flux, $N_i$) and presence of intrinsic noise. (A) Discriminable polarization angles versus R8 length fraction in the absence of intrinsic noise, without transduction unit saturation (solid curves) and with saturation (dashed curves). Four light intensities $N_i$; $1 \times 10^5$ (blue), $3 \times 10^5$ (red), $1 \times 10^6$ (green) and $3 \times 10^6$ (black) photons s$^{-1}$. (B) As in (A) but in the presence of intrinsic noise. Note the optimum division shifts to shorter R8 at higher intensities. (C) Optimum R8 length fractions from (B) versus incident photon flux without transduction unit saturation (solid curve) and with saturation (dashed curve). (D) As in (C), but R8 length optimizes mutual information. Degree of polarization $d = 0.1$.

polarization angles because photon noise dominates. As intensity increases further, intrinsic noise comes to dominate. The number of discriminable polarization angles is reduced and becomes less sensitive to changes in length fraction. Nonetheless there is still an optimum $\hat{l_8}$, which falls from 0.5 to 0.33 as $N_i$ increases from $1 \times 10^5$ to $3 \times 10^6$ photons s$^{-1}$ (Fig. 8C). This shortening of R8 is advantageous because when intrinsic noise dominates it is important to have a larger signal, and hence the higher *PS* of a shorter R8 (Fig. 6). Transduction unit saturation reduces the number of discriminable polarization angles above $N_i = 10^5$, and slightly reduces the optimum $\hat{l_8}$ (Fig. 8B and 8C).

We note in passing that our models identify three other factors that decrease the optimum $\hat{l_8}$ at highest light levels, albeit to much lesser extent (results not shown). These minor factors are increasing the dichroic ratio $\delta$, illuminating the CRP with monochromatic light at the peak absorption wavelength, and—up to a point—increasing the CRP length, $l$. All three factors allow R7 to act as a stronger filter, making it more worthwhile to increase $\hat{l_7}$ and reduce $\hat{l_8}$. The degree of polarization of clear blue sky is often much higher than 0.1. Increasing $d$ to 0.8 increases the number of discriminable angles but has little impact on optimum length fractions. We find that $\hat{l_8}$ is slightly reduced at the highest light levels.

In summary, with a CRP length $l = 100 \, \mu$m, the optimum division between R7 and R8 depends strongly on the relative contributions of photon noise and intrinsic noise, and weakly on transduction unit saturation. These factors are intensity dependent and favor

**Table 2  Fraction of the DRA CRP length occupied by R8 in different species of Diptera.**

| Species | $l_8/l$ | References |
|---|---|---|
| Rhagio scolopacea | 0.42 | Wada (1974a) |
| Leptempis | 0.45 | Wada (1974a) |
| Ceratitis capitata | 0.37 | Wada (1974a) |
| Drosophila melanogaster | 0.49 | Wada (1974a) |
| Scatophaga stercoraria | 0.57 | Wada (1974a) |
| Musca domestica | 0.53 | Wada (1974a) |
| Calliphora vicina | 0.4, 0.44 | Wada (1974a) and Wunderer & Smola (1982a) |
| Sarcophaga carnaria | 0.43 | Wada (1974a) |
| Zeuxia | 0.48 | Wada (1974a) |
| Lipoptena cervi | 0.54 | Wada (1974a) |

a shorter R8 at high intensities. Thus the optimum R8 length fraction, $\hat{l}_8$, reduces from 0.5 at lower light levels to 0.33 in bright light. This range is similar to the length fractions observed in the DRA's of different fly species; $\hat{l}_8 = 0.57$ to $\hat{l}_8 = 0.37$ (Table 2).

### Mutual information and length fractions of R7 and R8

We devised (Methods) a second measure of coding ability, mutual information, to confirm the conclusions drawn from numbers of discriminable polarization angles. Mutual information specifies the amount by which the opponent unit's output decreases our uncertainty about polarization angle, in bits (by definition a bit of information decides between one of two equally likely alternatives). Our derivation of mutual information assumes that the signal is obtained by integrating transduction rate over the integration time, that successive signals are independent, polarization angle has *a priori* a flat probability distribution, and the degree of polarization, is known to be constant. In our calculations, integration time $\tau = 90$ ms and degree of polarization, $d = 0.1$.

Mutual information depends on light intensity and the length fractions of R7 and R8 (results not plotted). At each intensity there is an R8 length fraction, $\hat{l}_8$, that maximizes the mutual information (Fig. 8D), which is almost indistinguishable from the $\hat{l}_8$ that maximizes the number of discriminable polarization angles (Fig. 8C). As expected, mutual information increases with light level. When coding with optimal length fraction at $N_i = 10^5$ photons s$^{-1}$, an opponent signal carries approximately 1.2 bits per integration time. Without saturation, mutual information increases with light level and approaches a ceiling of 1.7 bits per integration time, set by intrinsic noise. With saturation, mutual information increases with intensity to a maximum of around 1.5 bits per integration time and then decreases.

### CRP length and polarization coding

In the DRA the CRP formed by R7 and R8 is shorter, on two counts. First, a CRP in the DRA is approximately half the length of its neighbouring peripheral rhabdomeres, R1–6. Consequently the CRP stops well above the basement membrane, showing that there is plenty of space for a longer CRP. Second, CRPs in the DRA are $50 - 60\%$ shorter than CRPs in the rest of the eye; $l = 90{-}120\,\mu$m c.f. $240\,\mu$m (*Wada, 1974a*; *Wada, 1974b*; *Wunderer &*

*Smola, 1982a*). Could the pronounced shortening of CRPs in the DRA be a specialization for coding polarization?

To address this question, we used our optical and opponent models to see how two of our measures of performance, the opponent signal range and the number of discriminable polarization angles, change with CRP length, $l$. We modeled a range of lengths that exhibits all relevant effects, $l = 20$–$300\ \mu$m and, for simplicity, we kept the length fractions of R7 and R8 equal; i.e., $\hat{l_8} = \hat{l_7} = 0.5$.

### Opponent signal range and total length of the CRP

Without saturation, the opponent signal range, $\Delta Q$, reduces slightly with increasing $l$ (Fig. 9A), because self-screening decreases $PS$. Any increase in $PS_8$ due to stronger filtering by R7 is too weak to compensate for the loss in $PS_7$ due to self-screening because, as observed above, the relationship between $PS$ and $\Delta Q$ is non-linear, and $PS_8 > PS_7$. Nonetheless, the increase in $PS_8$ lessens the overall effect of increasing $l$. $\Delta Q$ falls by less than 10 percent as $l$ goes from 20 to 300 $\mu$m, irrespective of light intensity (Fig. 9B).

Saturation reduces signal range, $\Delta Q$, at all $l$'s, but the reduction becomes smaller as $l$ increases (Fig. 9A dashed curves). Saturation favors a longer CRP because as absorption decreases the photon flux along the rhabdomere the proportion of unsaturated microvilli increases. Thus it is advantageous to increase $l$ up to an optimum length, beyond which the benefits of relief from saturation are outweighed by losses from self-screening (Fig. 9A). The benefits of a longer $l$ increase with the severity of saturation, and hence with incident photon flux, $N_i$. Thus when $N_i = 1 \times 10^5$ photons s$^{-1}$ there is a barely perceptible optimum at $l = 60\ \mu$m. The optimum length increases to 120 $\mu$m at $3 \times 10^5$ photons s$^{-1}$ and 240 $\mu$m at $1 \times 10^6$ photons s$^{-1}$. With an incident flux $N_i = 3 \times 10^6$ photons s$^{-1}$ no optimum is a reached within the length range 20–300 $\mu$m. Note that although transduction unit saturation produces optimum lengths that depend on intensity, rhabdomeres that suffer less saturation have larger $\Delta Q$'s at all $l$'s. Thus a longitudinal pupil mechanism, which reduces saturation by attenuating rhabdomeric photon flux at high light levels (*Howard, Blakeslee & Laughlin, 1987*), will increase $\Delta Q$.

In summary, increasing $l$ at high intensities increases signal range $\Delta Q$ by reducing the loss in $PS$ produced by transduction unit saturation. This increase in $\Delta Q$ with $l$ follows the Law of Diminishing Returns, and is opposed by the effects of self-screening. At all but the highest intensity we modelled the CRP length used in the DRA, $l = 100\ \mu$m, performs close to optimum. Furthermore, extending the CRP to the length found in the rest of the eye is of little benefit. Even at the our highest light level, where extension to $l = 240\ \mu$m is most beneficial, the improvement in signal range is $< 8\%$.

### Number of discriminable polarization angles and total length of the CRP

Our second measure of performance, the number of discriminable polarization angles, almost invariably increases with total CRP length, $l$, according to the Law of Diminishing Returns. Both the magnitude of returns and the rate at which they diminish depend strongly on two factors—the relative effects of photon noise and intrinsic noise, and transduction unit saturation.

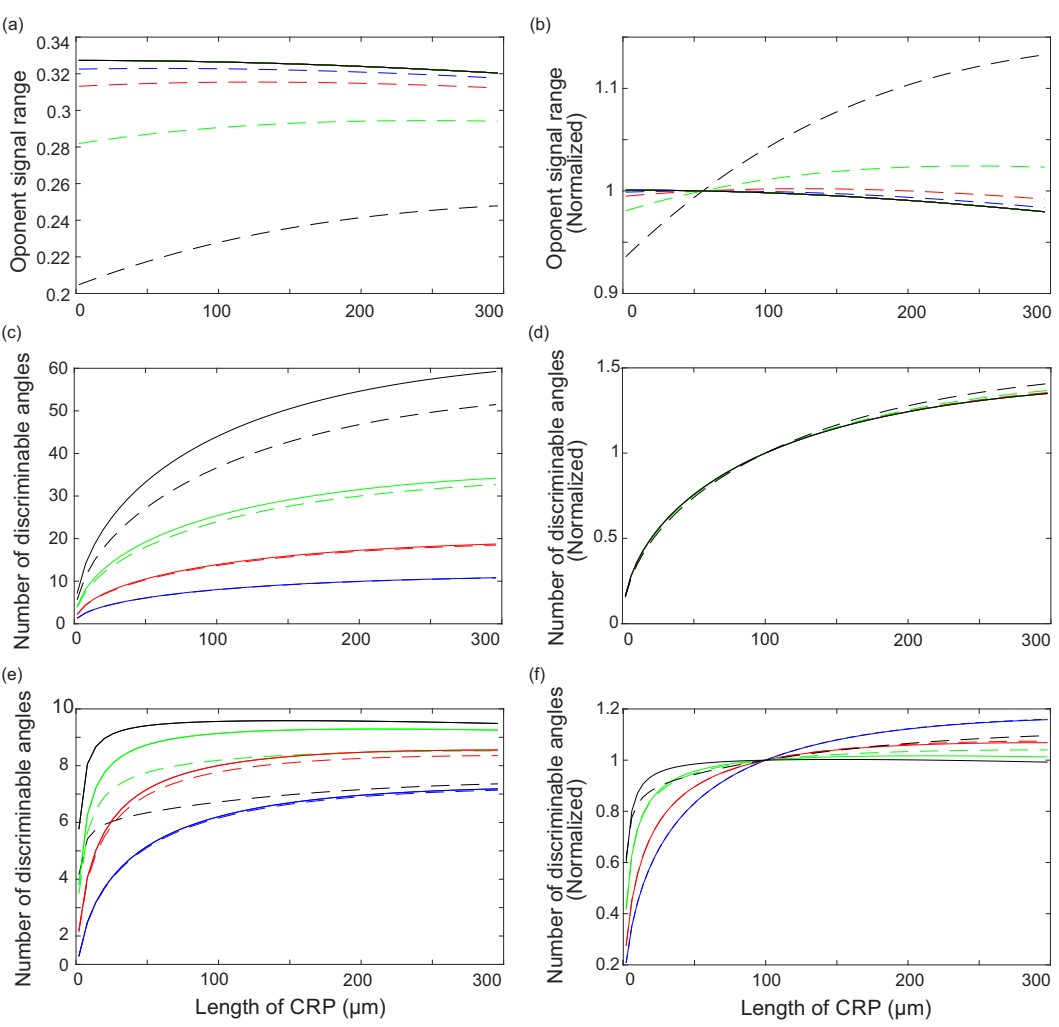

**Figure 9 Ability of opponent unit to code polarization depends of CRP length, *l*, according to relationships that depend on light intensity, transduction unit saturation and the presence of intrinsic noise.**
Most relationships follow the Law of Diminishing Returns. (A) Opponent unit signal range ($\Delta Q$) at four light intensities with transduction unit saturation (dashed) and without saturation (solid). Light intensities specified by incident photon flux, $N_i$ photons s$^{-1}$; $1 \times 10^5$ (blue), $3 \times 10^5$ (red), $1 \times 10^6$ (green) and $3 \times 10^6$ (black). (B) Curves plotted in (A), each normalised to its value at $l = 100 \,\mu$m. (C) Number of discriminable angles without intrinsic noise, $N_i$ and saturation state as in (A), and (D) curves normalized to $l = 100 \,\mu$m as in (B). (E) Number of discriminable angles as in (C), but with intrinsic noise $\sigma_{in}$. (F) Curves in (E) normalised to $l = 100$. Degree of polarization $d = 0.1$.

Without intrinsic noise, the number of discriminable polarization angles increases with length (Figs. 9C and 9D) according to the Law of Diminishing Returns. When the number of discriminable angles is normalized with respect to a CRP in the DRA, $l = 100 \,\mu$m, the increase in number with length is remarkably consistent; irrespective of intensity the number increases roughly 3-fold over our length range, along approximately the same curve (Figs. 9E and 9F, black lines). The number of discriminable angles improves because $SNR_7$ and $SNR_8$ increase with length (e.g., Fig. 3). The Law of Diminishing Returns is enforced by two non-linearities; $SNR$ increases as the square root of quantum catch (the

Square Root Law) and catch per unit length decreases exponentially with CRP length (Eq. (9)). Although the decrease in $\Delta Q$ with length (Fig. 9A) also diminishes returns, its contribution is minor. Transduction unit saturation slightly alters the relationship between length and normalized performance, by punishing the shorter CRPs and rewarding the longer ones (Fig. 9D).

Adding intrinsic noise changes the relationship between the number of discriminable polarization angles and CRP length, $l$, to a degree that depends upon the magnitude of photon noise, and hence quantum catch. When quantum catches are low, as happens at all $l$'s at lower intensities and with shorter $l$'s at higher intensities, photon noise dominates and intrinsic noise has almost no effect. In this situation the curves relating performance to length with intrinsic noise are virtually identical to those without (compare blue curves in Figs. 9E and 9C).

As quantum catch rises the effect of intrinsic noise increases and comes to dominate. The increasing effect of intrinsic noise is seen in the normalized plots (Figs. 9D and 9F). With no intrinsic noise, the number of discriminable polarization angles increases by 35% when $l$ is extended from 100 $\mu$m to 300 $\mu$m, independent of the light level. With intrinsic noise the growth with increasing $l$ is intensity dependent. At our lowest intensity ($N_i = 1 \times 10^5$ photons s$^{-1}$, blue curve) intrinsic noise has little effect in shorter CRPs, but it more than halves the increase produced when $l$ is extended from 100 $\mu$m to 300 $\mu$m, from 35% to 16%. At the next highest intensity ($N_i = 3 \times 10^5$ photons s$^{-1}$, red curve) intrinsic noise has a larger effect, particularly at longer $l$'s where quantum catch is higher. The increase in discriminable angles from $l = 100$ $\mu$m and $l = 300$ $\mu$m is cut 5-fold, from 35% to 7%. At $N_i = 1 \times 10^6$ photons s$^{-1}$ the increase is about 1%. At $N_i = 3 \times 10^6$ photons s$^{-1}$ there is almost no increase at all because, as shown in Fig. 9A, the curve is almost flat at $l = 100$ $\mu$m, and hits the upper limit imposed by intrinsic noise at $l = 150$ $\mu$m. To summarise, as quantum catch increases the effect of photon noise diminishes and the number of discriminable polarization angles approaches the upper limit set by intrinsic noise. When the incident photon flux is increased the limit is approached at shorter CRP lengths. Transduction unit saturation decreases the total number of distinguishable angles (Figs. 9C and 9E) both with and without intrinsic noise and, as explained above, favors longer CRPs (Figs. 9D and 9F) because a smaller proportion of microvilli are saturated.

### The DRA CRP has an efficient length

Our curves of performance (number of discriminable polarization angles) versus CRP length, $l$, allow us to evaluate the benefits of extending the CRP from the length found in the DRA, $l = 100$ $\mu$m to the length found in the rest of the eye, $l = 240$ $\mu$m. Elongation from 100 $\mu$m to 240 $\mu$m is of greatest benefit, 20%, at the lower light level $N_i = 1 \times 10^5$ photons s$^{-1}$. At higher light levels the benefit steadily reduces as photon noise becomes less important and at $N_i = 3 \times 10^6$ photons s$^{-1}$ there is no benefit without saturation. The CRP has hit the intrinsic noise ceiling. With saturation the benefit is <10%. Given that saturation happens, the best performance occurs at an incident photon flux, $N_i = 1 \times 10^6$ photons s$^{-1}$ (green dashed curve in Fig. 9E), in which case the benefit of extending from 100 $\mu$m to 240 $\mu$m is $\approx$5%. Thus shortening the CRP in the DRA increases the ratio between

discriminable angles and CRP length more than two-fold. On this basis, we conclude that the DRA's shorter CRP is a specialization that increases the efficiency with which a fly uses a sensor resource, rhabdomere length.

## DISCUSSION

In the dorsal rim area of the fly compound eye, the DRA, photoreceptors R7 and R8 are specialized to code the polarization of skylight. Because R7 and R8 form a central rhabdomere pair (CRP), with R7 placed in front of R8 (Fig. 1), R7 acts as a polarization filter that increases the polarization sensitivity of R8. We demonstrate how this tiered configuration sets up a trade-off between signal and noise. Lengthening R7, and hence increasing its absorption, increases R8's polarization sensitivity and hence R8's signal. However, lengthening R7 also reduces the number of photons R8 receives, thereby increasing the effect of photon noise. We evaluate this trade-off using a series of models, an optical model of photon absorption by R7 and R8, a model that accounts for the saturation of transduction units at high light levels, and an opponent model of polarization coding that introduces intrinsic noise. We find that with a CRP of fixed length, similar to that observed in the DRA, there are length fractions; i.e., divisions of the CRP between R7 and R8, that optimize polarization coding by maximizing signal to noise ratio (SNR). Saturation of transduction units at high intensities does not change these optimum length fractions, but reduces all measures of performance. Furthermore an optimal optical configuration, namely photoreceptor length fraction in a tiered CRP, depends in part on a neural factor, the level of intrinsic noise.

The intensity dependent range of optimum length fractions, 0.5–0.33, matches the range of length fractions observed among flies (Table 2), suggesting that R7 and R8 divide a resource, a CRP of given length, to optimize their ability to code polarization. Additional evidence for efficient resource allocation is obtained by noting that in the DRA R7 and R8 make a CRP that is shorter than the peripheral rhabdomeres of photoreceptors R1–6, and 50%–60% shorter than CRPs in the rest of the eye. Our models show that this reduction in length increases the efficiency with which the DRA R7/R8 uses CRP to code polarization; it doubles the ratio between quantitative measures of coding ability and CRP length.

We will now discuss the procedures we used, the validity of their assumptions, their relationship to previous studies, and the novelty of their contributions. We will close by assessing the impact of our findings on our understanding of the structure, function and design of photoreceptor arrays.

### Modelling optical absorption

Our absorption model quantifies the optical trade-off between signal and noise by estimating the rates at which the visual pigment molecules of photoreceptors R7 and R8 absorb photons when viewing a small patch of polarized skylight. Our model confirms that optical effects within a CRP, namely filtering and self-screening, play important roles in determining polarization sensitivity, as first demonstrated in the CRP of fly (*Snyder, 1973*; *Gribakin & Govardovskii, 1975*). We also confirm that, as shown later in fused rhabdoms (*Nilsson, Labhart & Meyer, 1987*) and banded rhabdoms (*Stowe, 1983*), a photoreceptor's

PS depends on its length, the orientation of its microvilli, the percentage of the rhabdom's (or CRP's) microvilli it contributes, and the percentages and orientations of the microvilli contributed by the photoreceptors that screen it. In other tiered structures filtering by a distal photoreceptor also sharpens and repositions the spectral absorption peaks of a proximal photoreceptor, as demonstrated in the rhabdoms of butterflies (*Stavenga & Arikawa, 2006*) and stomatopods (*Marshall, Cronin & Kleinlogel, 2007*). However ours is the first study to investigate a trade-off between signal and noise, set up by optical interactions within a CRP or fused rhabdom.

The trade-off between signal and noise set up by filtering has been demonstrated and analyzed in those cone photoreceptors that place a coloured oil droplet in front of their visual pigment; i.e., in the cone inner segment, between the entrance aperture for light and the outer segment (*Vorobyev et al., 1998*). The oil droplet sharpens the cone's spectral sensitivity by filtering the light delivered to the visual pigment, but also increases the effect of photon noise by reducing quantum catch. Our study of this optical trade-off is distinctive in two ways. The optical filter we consider is also a photoreceptor, R7, and this photoreceptor operates in tandem with the photoreceptor it shields, R8, to code polarization.

To model absorption rates we make several simplifying assumptions. We assume that the dichroism produced by the alignment of rhodopsin molecules in microvilli is the only effect that changes the polarization of light as it travels down a rhabdomere. In fact, the rhabdomere is also optically anisotropic. However the resulting birefringence of a fly rhabdomere, $\Delta n < 1.0 \times 10^{-3}$ (*Kirschfeld & Snyder, 1975*; *Beersma et al., 1982*), produces a very small phase advance

$$\Delta \varphi = \frac{2\pi l \Delta n}{\lambda} \qquad (39)$$

which in a rhabdomere of length $l = 60\ \mu m$ is $<1.13$ rad. Because this phase advance is equivalent to $<18\%$ of a wavelength, it will have little effect on R7/R8 rhabdomeres in the DRA. Furthermore, optical experiments measured negligible birefringence in the longer DRA rhabdomeres of ants (75–85 $\mu m$) and crickets (150–200 $\mu m$) (*Nilsson, Labhart & Meyer, 1987*). In even longer rhabdomeres, birefringence can become important, producing mode beating between the polarized modes, as found in some butterflies (*Nilsson, Land & Howard, 1988*). This beating reduces PS with increasing length, thereby accentuating the Law of Diminishing Returns that characterises the relationship between the coding ability of R7/R8 and CRP length (Fig. 9).

We calculated absorption at different wavelengths using a popular template for the spectral sensitivity of rhodopsins (*Stavenga, Smits & Hoenders, 1993*). This template provides an acceptable approximation of the absorption curve of a UV rhodopsin, although it slightly overestimates the curve's width (*Stavenga, 2010*). We disregard absorption by metarhodopsin because it absorbs in the blue, whereas almost all of the photons absorbed by the UV rhodopsin are at wavelengths below 413 nm (Fig. 2). Moreover, because daylight delivers more photons in the blue and the green, the rhodopsin:metarhodopsin ratio

remains high in bright light. Consequently we can safely ignore the loss of sensitivity due to rhodopsin depletion.

We had to calculate the absorption of polarized skylight by UV rhodopsin for a large number of combinations of rhabdomere length. To do this quickly and efficiently, we used the approach introduced by *Warrant & Nilsson (1998)*. We fitted a convenient non-parametric function to a much smaller number of exact calculations of absorption, each made at a different length by integrating across all relevant wavelength. Although we use a different non-parametric function, the calculations we make agree well with calculations made using Warrant and Nilsson's function for most photoreceptor lengths (Fig. 3), with one small exception. Their function breaks down at very short lengths, where PS should equal the dichroic ratio. *Alkaladi, How & Zeil (2013)* adapted Warrant and Nilsson's approximation to analyse the PS of banded rhabdoms of the fiddler crab, and arrived at yet another expression, which is incompatible with the general expression for the absorption of polarized light in a tiered system (Eqs. (11) and (13)). The mismatch arises because their derivation implicitly assumes that the light leaving each band has the same wavelength content as the light entering the photoreceptor from the facet lens.

The spectral composition of skylight changes according to the time of day, especially at dawn and dusk. We did not take this into account because changes in the relative contributions of wavelengths within the absorption band of the CRP's UV rhodopsin are small. The large changes are in the relative contributions of wavelengths below and above 450 nm (*Johnsen et al., 2006*).

Our findings rely on assumptions of homogeneity; namely that R7 and R8 have identical properties, aside from length and microvillar orientation, and the rhabdomere's diameter and optical properties do not change with length. The limited amount of physiological (*Hardie, 1984*) and anatomical (*Wada, 1974a*; *Wada, 1974b*; *Wunderer & Smola, 1982a*) data does not indicate otherwise, with the exception that the cross-sectional area of R7's rhabdomere is on average 26% less than R8's in *Calliphora vicina* (*Wunderer & Smola, 1982a*).

There is no experimental data supporting the trade-off evaluated by our optical model. There are no published measurements of photoreceptor noise in the fly DRA and the prediction that R8 has a higher PS than R7 has not been reported (*Hardie, 1984*). Nonetheless, there are three reasons why the failure to find a higher PS does not necessarily invalidate our model. The first is the small number of intracellular recordings from R8 photoreceptors in the DRA. The second is the difficulty of making reliable recordings and measurements from such small cells. The third is that when PS was measured it was done using 365 nm light (*Hardie, 1984*). At this wavelength the UV rhodopsin's absorption is half maximal, which reduces the effect of filtering by R7 on R8.

Our most contentious assumption is that we ignore absorption by the longitudinal pupil; the pigment granules that a fly photoreceptor moves close to its rhabdomere to attenuate bright light (*Franceschini & Kirschfeld, 1976*; *Stavenga, 1979*). Attenuation by a pupil has not been measured in the DRA, but in other eye regions the pupil "closes" progressively at high intensites to attenuate the rhabdomeric photon flux in R1–6 photoreceptors by up to 2 log units. To a first approximation pupil attenuation will be equivalent to reducing

the incident photon flux, $N_i$. Although the pupil preferentially absorbs longer wavelengths (*Stavenga, 2004*) this will have a small effect on the spectral sensitivities of photoreceptors R7 and R8 in the DRA, because they absorb with a UV pigment. In any case, the pupil's action will not alter the trade-offs and optimal length fractions we report. The pupil operates at high intensities to extend the range of incident photon fluxes at which the trade-offs and optima occur. It is at these high intensities that transduction unit saturation can change the relationship between photons absorbed and photons transduced to electrical signals, as discussed in the next section.

## From photons absorbed to photons transduced

We converted photoreceptor absorption rates to transduction rates because the information photoreceptors code depends upon the numbers of photons they transduce to electrical signals. Measurements of single photon responses (quantum bumps) show that a single fly photoreceptor (type R1–6) transduces more than 50% of photons that arrive at the lens, within its acceptance angle (*Dubs, Laughlin & Srinivasan, 1981*). This means that well over 50% of the photons incident on its rhabdomere are transduced so, for simplicity, we assume a quantum efficiency of 1. In this case, the transduction rate equals the absorption rate. Noise analysis shows that a fly R1–6 photoreceptor maintains its high quantum efficiency up to transduction rates of $10^4$ photons s$^{-1}$ (*van Steveninck & Laughlin, 1996a*). Above this intensity, quantum efficiency falls due to the action of the longitudinal pupil (discussed above) and the saturation of transduction units (*Howard, Blakeslee & Laughlin, 1987*).

To account for the effects of transduction unit saturation on the mean and variance of transduction rate, we extended the original binomial model (*Howard, Blakeslee & Laughlin, 1987*). Our improved model confirms that saturation had no effect on quantum efficiency, and hence signal and noise, below rates of $10^4$ photons s$^{-1}$ and that above this it progressively reduces signal and signal to noise ratio.

Basic binomial models simplify the causes and effects of transduction saturation. Do these simplifications invalidate our major conclusions? The binomial model assumes a dead time (refractory period) of 30 ms. The dead time has not been determined directly, however detailed, well-informed simulations of the effects of transduction unit saturation (*Song et al., 2012*; *Song & Juusola, 2014*) suggest a range of mean values, from 10 ms to 100 ms, depending on fly species. Our value, 30 ms, is towards the lower end of this range. Changing this value will not change the nature of the trade-offs, it will simply shift the intensity range of saturation effects, according to the inverse of dead time.

A second simplification is used to calculate signal and noise. We sum transduction events (quantum bumps) over a 90 ms time interval. However signals, absorption rates, quantum bump latencies, and dead times vary continuously over time. Consequently the effects of transduction on signal and noise depend upon the time varying properties of signals and photoreceptor response dynamics. The detailed simulations encapsulate these dynamics and show how they improve the coding of stimuli with natural dynamics (*Song et al., 2012*; *Song & Juusola, 2014*). These dynamic effects may well play a role in coding skylight polarization but, in the absence of measurements of the natural time series of intensities experienced by R7 and R8 in the DRA, a detailed simulation seems premature.

Moreover incorporating the dynamics of signals, signalling and saturation is unlikely to overturn our major conclusion that, given an optical signal, photon noise and intrinsic noise there is an optimal division of CRP length between R7 and R8. The optical trade-off between signal and noise is based on fundamental principles and is little affected by transduction unit saturation. The beneficial effects on coding documented by the more complicated models are largely due to changes in quantum efficiency associated with large bursty contrast changes. They are not forthcoming in responses to stimuli that lack more prolonged dark contrasts (*Song & Juusola, 2014*), which may well be the stimuli coded by R7 and R8, because polarization and brightness change gradually and by relatively small amounts across a bright blue sky. The more complicated effects of saturation might come into play when the DRA of a rapidly turning fly views the sky through a broken canopy of vegetation.

Finally, both our model and the detailed simulations (*Song et al., 2012*; *Song & Juusola, 2014*) set aside the action of the longitudinal pupil. The pupil progressively attenuates the rhabdomeric photon flux to keep the transduction rate close to an experimentally demonstrated optimum, a peak in the contrast SNR that is captured by the binomial model (*Howard, Blakeslee & Laughlin, 1987*). Our results concur (Fig. 8). As intensity increases past the point at which saturation cuts in, discriminability continues to increase, but at a declining rate. Discriminability then peaks and falls by as much as 25% in brightest sunlight. This observation shows that a longitudinal pupil in the DRA's R7/R8 could usefully play the role advocated by *Howard, Blakeslee & Laughlin (1987)*; limiting saturation so as to operate close to peak SNR. To determine the extent to which tranduction unit saturation changes the ability of a dorsal rim R7/R8 pair to code polarization, we must account for the pupil.

## Opponent coding and discriminability

The use of an opponent model can be justified on several grounds (*Hempel de Ibarra, Vorobyev & Menzel, 2014*). First, neural processing involves opponent mechanisms. In color vision opponent mechanisms operate at the first stages of processing and at higher levels, in both insects and vertebrates. In the polarization pathways served by DRAs there is strong evidence from *Drosophila* for opponent interactions between the output terminals of R7 and R8 (*Weir et al., 2016*), and opponent interactions are observed in neurons at higher levels (*Labhart, 1988*; *Labhart, 2000*; *Labhart, Petzold & Helbling, 2001*; *Vitzthum, Müller & Homberg, 2002*). Second, opponent processing provides a simple way to code a sub-modality, like color or polarization, independent of background intensity. Third, spatially and spectrally opponent mechanisms eliminate redundancy (*Srinivasan, Laughlin & Dubs, 1982*; *Buchsbaum & Gottschalk, 1983*). Fourth, and most importantly for our purposes, opponent models of color vision account for exacting behavioral measures of discrimination (*Hempel de Ibarra, Vorobyev & Menzel, 2014*).

As in studies of color vision, we assess discriminability using an opponent model that scales and combines signals and noise (*Hempel de Ibarra, Vorobyev & Menzel, 2014*). An opponent unit takes the difference between the inputs from R7 and R8, scaled to represent contrast by dividing photon rates and counts by their means. R7 and R8's scaled signals

subtract and, being uncorrelated, their scaled noise variances add. Then intrinsic noise is added to generate an opponent output. This intrinsic noise has a constant variance, independent of light level. The opponent output represents a combination of polarization angle and degree of polarization, and noise determines just noticeable differences in this combination. The jnd's define discriminability and the relationship between opponent signal and noise determines mutual information.

*How & Marshall (2014)* were the first to apply this approach to polarization vision. An earlier model (*Nilsson, Labhart & Meyer, 1987*) showed how a polarization-opponent signal is affected by an unequal share of microvilli directions in a fused rhabdom, but did not include noise. How and Marshall used their opponent model to show how under natural conditions the rhabdomeres in the fiddler crab are optimally oriented to detect the polarization degree rather than the polarization angle. To set the noise level they assumed a single source and adjusted its variance to account for jnd's in polarization, measured behaviourally. We independently applied their approach, and in doing so we extended it to separate the effects of intensity dependent photoreceptor noise (e.g., photon noise) and intrinsic noise.

Our estimates of discriminable angles and mutual information depend upon assumptions about integration time and intrinsic noise variance. The chosen integration time interval, $\tau = 90$ ms, is arbitrary. It will affect the absolute values of SNR, numbers of angles and mutual information but not their relative values (e.g., Eq. (19)). Thus the shapes of curves relating performance to length and light level are the same and the optima are unchanged.

There are no measurements of intrinsic noise in the neural pathways served by the DRA so we chose an intrinsic noise variance ($\sigma_{in}^2 = 5 \times 10^{-5}$) that is about 7.5 times greater than photon noise at an incident photon flux of $N_i = 10^6$ photons s$^{-1}$. This value is in accordance with measurements of noise in large monopolar cells (LMCs), directly post-synaptic to photoreceptors R1–6 (*van Steveninck & Laughlin, 1996b*). At medium frequencies, around 50 Hz, an LMC's intrinsic noise is 7 times that fed through from photoreceptors R1–6 and at high frequencies, around 100 Hz 10 times greater. Because the polarization signal changes gradually across the sky we favoured the lower value. The R1–6 axon terminals in the lamina form a massive parallel array with 1200 synaptic vesicle release sites driving each LMC (*Nicol & Meinertzhagen, 1982*; *van Steveninck & Laughlin, 1996b*). The DRA's R7 and R8 have large output terminals in the medulla. However, the numbers of vesicle release sites and release rates have not be determined, so we may well have underestimated intrinsic noise.

The underestimate will not be huge, nor will it change greatly the trade-offs we describe and the optima they create. An increase in intrinsic noise simply reduces the light intensity at which it starts to dominate photon noise, and this shifts the curve relating optimum R8 length fraction to light intensity to lower intensities. For example, when the number of release sites is reduced by 90% the intrinsic noise variance is ten times larger ($\sigma_{in}^2 = 5 \times 10^{-4}$). Using this value in our opponent model, we calculate that the optimum length fraction of R8 at $N_i = 10^6$ photons s$^{-1}$ drops to 0.25 and the Law of Diminishing Returns on CRP length becomes more severe because performance approaches a lower intrinsic noise ceiling

(results not plotted). In other words, higher levels of intrinsic noise favour shorter CRPs because they use the sensor resource, CRP length, more efficiently. We will return to this point in our final section.

## Coding polarization signals from the dorsal rim area

No matter how they are processed, the signals coded by a pair of R7 and R8 cannot give unequivocal information about the polarization state of incoming light. Polarization angle is confounded by degree of polarization (*Bernard & Wehner, 1977*). Orientation cues are obtained from sky polarization patterns by integrating information from many CRPs, each sampling a different patch of sky. Evidence from other insects, such as crickets, shows that higher order POL neurons integrate information from across the DRA, and suggests that they receive this information from small field opponent units, similar to the ones we model (*Labhart, 1988*; *Labhart, Petzold & Helbling, 2001*). Given the advantages of such an elementary opponent unit (see above), using it for the first stage in neural processing makes sense.

Our models could help us to understand how low intensity polarization patterns are resolved because they evaluate the effect of photon noise. This might help to explain how the DRA's of nocturnal beetles, and the neural pathways they serve, are adapted to support a remarkable behaviour—orientation to the polarization pattern of the night sky (*El Jundi et al., 2015*). Our models evaluate the role of optics in determining ratios between polarization signal and photon noise and translate signal and noise into discrimination thresholds and mutual information, and this allows one to search for optimum sampling and processing strategies. Our modelling also takes account of the relative contributions of photon and intrinsic noise, which may be important because, when photoreceptor signals are pooled neurally, the ratio between photon and intrinsic noise will decrease. In other words, we have done in theory what beetles may have done in practice, optimized the inevitable trade-off between polarization signal and noise within a constraint imposed by limited resources, in our case rhabdomere length.

## The efficient use of a sensor resource, rhabdomere length

Our study shows that photoreceptors R7 and R8 in the DRA are adapted to make efficient use of rhabdomere length. There are good reasons why both the length of the CRP and lengths of its constituent rhabdomeres should be allocated efficiently; length represents three limiting resources, materials, space and metabolic energy consumption, as follows. For a photoreceptor of given cross section, the consumption of space and materials increases in direct proportion to length. The relationship between energy consumption and length is less direct, but compelling. For a rhabdomere of given cross-section, the number of microvilli increases in proportion to length. More microvilli means a larger light-gated conductance, and to avoid saturation of membrane potential, a larger potassium conductance. These larger conductances carry larger currents which consume more energy (*Howard, Blakeslee & Laughlin, 1987*), as demonstrated by comparing photoreceptors of different length (*Niven, Anderson & Laughlin, 2007*). This length-dependent energy consumption is significant; direct measurements confirm that retinal photoreceptors

account for 8% of a blowfly's resting oxygen consumption (*Pangršič et al., 2005*). Given such a high level of consumption, adaptations that improve photoreceptor energy efficiency should promote fitness. On these grounds we suggest that it is valid to treat rhabdomere length as a limiting resource, to be used efficiently.

We demonstrate two routes to efficiency. One is to divide the length of the CRP between photoreceptors R7 and R8 so as to optimize two measures of coding ability, number of discriminable polarization angles and mutual information. The division made in the fly DRA is close to optimal, and there are reasons to suggest that CRPs in the rest of the eye are divided likewise. In the rest of the eye there are spectrally distinct classes of CRP, suitable for color vision (*Wunderer & Smola, 1982b*; *Hardie, 1985*). In *Calliphora vicina*, there are two spectral classes of CRP, R7y/R8y and R7p/R8p, distributed randomly across the photoreceptor array. There are also two morphological classes of R8 with different relative lengths that, on the basis of their relative frequencies, can be associated with the two spectral classes (*Smola & Meffert, 1979*; *Wunderer & Smola, 1982b*). In the light of our modelling of the DRA, this evidence suggests that the length fractions of R7 and R8 are allocated to increase the efficiency of color coding. There are two reasons why the R7p/R8p CRP could benefit from a shorter R8, and hence longer R7. One is that, because the spectral sensitivity curves of R7p and R8p overlap more than the curves of R7y and R8y, the signal in R8p will benefit more from the sharpening of spectral sensitivity produced by a longer R7p. The other is that, because the spectral sensitivity curve of R7p is narrower than the curve of R7y, and centred in shorter wavelengths where fewer photons are available, a longer R7p is needed to combat photon noise. Our model of polarization opponency could obviously be adapted back to color opponency to test these hypotheses, but the results may well depend on assumptions about what a fly uses its spectral classes of photoreceptors for.

The second route to efficiency is to regulate CRP length. Our modelling shows how efficient usage of CRP length can explain why the CRP in DRA is approximately half the length of both its neighbouring peripheral rhabdomeres (R1–6) and CRPs elsewhere in the eye. The DRA's shorter CRPs are more efficient for coding polarization because the relationship between CRP length and number of discriminable polarization angles follows the Law of Diminishing Returns (Fig. 9). This version of the law is seemingly inescapable because it is enforced by biophysical constraints on signal and noise. Quantum catch increases sub-linearly with rhabdomere length due to exponential absorption. (Eq. (9), Fig. 3) and at lower light levels, SNR increases as the square root of quantum catch. At the highest light levels, another length dependent factor, number of available transduction units (i.e., microvilli), constrains signal and noise, and the maximum achievable SNR tends to increase as the square root of the total number of transduction units (*Howard, Blakeslee & Laughlin, 1987*). Constrained by these factors, doubling the length of the DRA's CRP to equal that in the rest of eye increases the number of discriminable polarization angles by less than 10%, but halves the efficiency with which R7 and R8 use CRP length to code polarization.

Our theoretical arguments add to a growing body of experimental evidence that the number of transduction units, and hence microvilli, are a limiting resource to be employed efficiently. This evidence has been obtained by comparing photoreceptors within a single

retina, and in the retinas of different species. Within the blowfly retina, photoreceptors R7 and R1–6 increase their SNR's with light intensity, and in full daylight approach a maximum asymptotically. This maximum SNR is lower in R7 than in R1–6, in accordance with there being fewer microvilli in R7's shorter rhabdomere (*Anderson & Laughlin, 2000*). Comparing R1–6 in the same retina, SNRs in bright light and information rates correlate with the optical resolving power of the photoreceptor array, being higher in the frontal eye region, which samples more densely with narrower acceptance angles (*Burton, Tatler & Laughlin, 2001*). This suggests that transduction units are allocated according to need.

Turning to comparisons among species, measurements made from homologous R1–6 photoreceptors in four species of fly of increasing size, show that SNRs, information rates and energy consumption are higher in longer photoreceptors, while efficiencies (bits coded per ATP consumed) are lower, according to the law of diminishing returns (*Niven, Anderson & Laughlin, 2007*). Comparing two species of similar size, the predatory killer fly *Coenesia* supports faster and more acute vision than the fructivorous *Drosophila*. To support its behaviour *Coenesia* uses longer photoreceptors with more microvilli to achieve a higher SNR (*Gonzalez-Bellido, Wardill & Juusola, 2011*). Thus comparative studies argue strongly that photoreceptor length is a limiting resource that is applied efficiently, according to behavioural requirements and constraints imposed by biophysics and the properties of natural signals (*Sterling & Laughlin, 2015*, Chapter 8).

## Conclusion and outlook

Our study of the optimal allocation of photoreceptor length for opponent polarization coding confirms that having longer rhabdomeres with more microvilli improves vision by increasing the SNR at a given rate of incident photon flux. This increase is achieved in two ways. At low light levels increasing microvilli increases quantum catch, and in full daylight it increases the rate at which transduction units register photons. Thus the number of microvilli plays an important role in the function, design and evolution of compound eyes (*Howard & Snyder, 1983*). Future studies of the ways in which the optics of compound eyes, and especially tiered CRPs, are adapted to visual ecology should take account of the limitations on signal and noise imposed by numbers of microvilli and their lengths (*Gonzalez-Bellido, Wardill & Juusola, 2011*). Our study shows how this can be done with an opponent model. We suggest that a fertile new approach, namely modifying opponent coding models from colour vision in order to relate the coding and processing of polarization signals to behaviour (*How & Marshall, 2014*), could well bear even more fruit if, like our study and the colour opponent models that have gone before, one takes into account intensity dependent photoreceptor noise. It will be interesting to see if future applications confirm our finding that the optimal configuration of a photoreceptor array depends not just on optics and transduction rate, but on the intrinsic noise introduced during neural transmission and processing. If this proves to be the case, efficiency will be improved by matching the allocation of resources in one component of a system, optics, to the resources invested in other components, neural processing. This design strategy, matching the application of resources to components within a system to optimize overall efficiency, is commonly observed in neurons and neural circuits (*Sterling & Laughlin,*

*2015*). Here we have demonstrated how this design strategy is implemented within a CRP, by allocating a sensor resource, rhabdomere length, to R7 and R8.

## ACKNOWLEDGEMENTS

The authors thank D.G. Stavenga for peerless advice on optics and presentation, and R.C. Hardie, K.D. Longden, M. Lengyel, J.E. Niven, and M. Wicklein for discussions and feedback on previous versions of this manuscript.

### Funding

Simon Laughlin is supported by an honorarium from H. Britton Sanderford. Francisco Heras was supported by grants from Fundación Caja, Madrid; Trinity College, Cambridge; and the Departmentt of Zoology, University of Cambridge. The funders had no role in study design, data collection and analysis, decision to publish, or preparation of the manuscript.

### Grant Disclosures

The following grant information was disclosed by the authors:
H. Britton Sanderford.
Fundación Caja, Madrid.

### Competing Interests

The authors declare there are no competing interests.

### Author Contributions

- Francisco J.H. Heras conceived and designed the experiments, performed the experiments, analyzed the data, contributed reagents/materials/analysis tools, wrote the paper, prepared figures and/or tables, reviewed drafts of the paper.
- Simon B. Laughlin conceived and designed the experiments, wrote the paper, reviewed drafts of the paper.

### Data Availability

GitLab: https://gitlab.com/fjhheras/optimizing_pol_2016.

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
