# Peer review of "Optimizing the use of a sensor resource for opponent polarization coding"

_PeerJ, doi:10.7717/peerj.2772_

## Round 0.1 · original submission · Minor Revisions

· Academic Editor

Minor Revisions

With the two positive feedbacks from the reviewers, I am happy to accept your article, provided the suggestions of the two reviewers are incorporated and/or considered.

·

Basic reporting

No comments

Experimental design

Excellent and timely. A very worthwhile extension of current thought in the area.

Validity of the findings

Both very interesting and very valid - see general comments below for more detail.

Comments for the author

Very nice manuscript! Apologies for not getting back sooner – a few flooding issues in lab.
This is an exhaustive and exhausting manuscript to digest. It certainly investigates the idea and central hypothesis from every angle – all 360 of them!

Abstract
Define ‘mutual information or to start use a different term? Otherwise we have to skip to line 233 to find it.

Fig. 1
Add an arrow and “Receptor noise” label perhaps? At the moment it seems the only noise is after the interneuron? Maybe?

Introduction
At the end of the third paragraph, explain briefly why R7&8 can’t get any longer? Is there something in the way? Why are they not as long as R1-6? The answer becomes clear from your modelling later but is there a mechanical / packing reason too?

Methods
Why use Kok’s 1972 light measurement. There are more recent and possibly better measurements available now? Probably makes little difference to any of the conclusions mind you!

I the less than 100% polarisation nature of skylight taken into account? Absorption rate calculations? How does the model perform at very low degree differences? Degree being constant is mentioned at line 241 but what if – as is often the case in real sky, the actual degree of pol is say 40% only?

Thought in passing – are the microvilli of R7 and R8 of identical diameter; is an assumption of equal dichroic ratio valid?

Discussion
While some mention is made of the absorption of lateral pigment, did the calculations also consider different wave-guide modes and how much light travels in the photoreceptors based on suitable mode fractions? What would happen with a slight or indeed doubling of rhabdomeric diameter?
The idea of allocating transduction units (microvillar number or perhaps dimension – diameter and packing) according to need is a very interesting one. Although possibly outside the scope here – were there any considerations given to other potential dimensions? Aspect-ratio of rhabdomeric cross section, diameter or shape of rhabdomeric cross section? Could the model be asked to find the ideal orthogonal polarisation ‘unit’. This feeds questions such as why are photoreceptors associated with PS often square in cross section? Is there more to this shape than meets the eye? (Sorry!).

·

Basic reporting

Language: Clear and well-written.
Introduction and background: Justification for the paper is clearly established and well-grounded in current research.
Citations: Relevant and comprehensive.
Structure: Closer linkings between methods and results could improve procedural clarity.
Figures: Readable, some slightly difficult to parse in b&w.

Experimental design

Research is original and well-defined, methods and procedure are outlined rigorously. Contributes substantially to the literature.

Validity of the findings

Research is within journal scope, conclusions are well-supported.

Comments for the author

A thorough and detailed model of the polarization-sensitive tiered fly rhabdom is developed here - an extremely welcome and timely addition to the literature. Core assumptions and simplifications made are largely well-justified, and the chosen parameters are well-sourced from existing research.

My only major suggestion is that the discussion section would benefit from a closer examination of the ethological context of different fly species, as a way to verify or validate the claim of maximal efficiency in the physical structure of the photoreceptors. Particularly considering the range of length ratios measured by Wada, and the role incoming light level plays in determining the optimal length ratio between the two photoreceptors, this could serve to further test or validate the model.

Specific comments
Line 1 (title): Title could be better representative of the content - “a sensor resource” is somewhat vague and uninformative.

Line 77: Incoming spectral radiance is calculated for a cloudless sky at high solar position. Spectral composition changes dramatically according to time of day (and according to solar position relative to viewing angle). However it is not apparent how this link between incoming light intensity and wavelength composition is accommodated in the model. A non-monochromatic model of absorptance is established in the methods, then later (line 319) the claim is made that length ratios for PS maximization are not dependent on wavelength composition, however the results demonstrating this are not clearly shown (or at least are not obvious to this reviewer!).

Figure 2: Source for absorption function (solid line)? Stavenga?

Line 90: Showing an intermediate substitution may help the reader parse the derivation here

Line 176 (see also 257): The assumption of small contrast signals is valid for low polarization degree (d), and when generating their results, the authors have used a close-to-threshold value of d=0.1. However (as is pointed out by the authors), the polarization degree of unclouded skies is frequently much higher than 0.1. This will affect the SNR assumptions to a certain extent, and any corresponding effects on the ultimate optimal length ratios should ideally be explored, if possible.

(line 313: typo, Figure 4i should be Figure 4)

Line 432: Introduction of an acronym (jnd) which is not explained until later in the paper (line 729) - very confusing. Also since the acronym is used only twice, it is probably unnecessary.

---

## Round 0.2 · accepted · Accept

· Academic Editor

Accept

Articles such as your make it a pleasure to be an editor. Thanks for submitting it to us!

·

Basic reporting

No comments

Experimental design

No comments

Validity of the findings

No comments

Comments for the author

Revision looks great - all points and suggestions accounted for.

·

Basic reporting

No comments

Experimental design

No comments

Validity of the findings

No comments

Comments for the author

Having now read the rebuttal and the revised manuscript, I'm satisfied with the author's responses - the updates to the text and equations have significantly improved the clarity of the results and analysis, and my remaining suggestions/reservations were satisfactorily addressed by the authors in their comments. I'm still not entirely convinced about the title, but only because I want to be sure the research gets the wide audience it deserves - I will leave this to the discretion of the authors!
I greatly look forward to the publication of this excellent and thorough paper.